

# The hierarchy of controls on snowmelt-runoff generation over seasonally-frozen hillslopes

Anna E. Coles[1], Willemijn M. Appels[1,2], Brian G. McConkey[3], and Jeffrey J. McDonnell[1,4]

[1] Global Institute for Water Security & School of Environment and Sustainability, University of Saskatchewan, 11 Innovation Boulevard, Saskatoon, SK S7N 3H5, Canada
[2] Lethbridge College, 3000 College Drive S, Lethbridge, AB T1K 1L6, Canada
[3] Swift Current Research and Development Centre, Agriculture and Agri-Food Canada, PO Box 1030, Swift Current, SK S9H 3X2, Canada
[4] School of Geosciences, University of Aberdeen, Aberdeen, AB24 3UE, United Kingdom

*Correspondence to:* Anna E. Coles (anna.coles@usask.ca)

**Abstract.** Understanding and modeling snowmelt-runoff generation in seasonally-frozen regions is a major challenge in hydrology. Partly, this is because the controls on hillslope-scale snowmelt-runoff generation are potentially extensive and their hierarchy is poorly understood. Understanding the relative importance of controls (*e.g.* topography, vegetation, land use, soil characteristics, and precipitation dynamics) on runoff response is necessary for model development, spatial extrapolation, and runoff classification schemes. Multiple interacting process controls, the nonlinearities between them, and the resultant threshold-like activation of runoff, typically are not observable in short-term experiments or single-season field studies. Therefore, long-term datasets and analyses are needed. Here, we use a 52-year dataset of runoff, precipitation, soil water content, snow cover, and meteorological data from three monitored *c.*5 ha hillslopes on the Canadian Prairies to determine the controls on snowmelt-runoff, their time-varying hierarchy, and the interactions between the controls. We use decision tree learning to extract information from the dataset on the controls on runoff ratio. Our analysis shows that there was a variable relationship between total spring runoff amount and either winter snowfall amount or snow cover water equivalent. Other factors came into play to control the fraction of precipitated water that infiltrated into the frozen ground. In descending order of importance, these were: total snowfall, snow cover, fall soil surface water content, melt rate, melt season length, and fall soil profile water content. While mid-winter warm periods in some years likely increased soil water content and/or led to development of impermeable ice lenses that affected the runoff response, hillslope memory of fall soil moisture conditions played a strong role in the spring runoff response. The hierarchy of these controls was condition-dependent, with the biggest differences between high and low snow cover seasons, and wet and dry fall soil moisture conditions. For example, when snow cover was high, the top three controls on runoff ratio matched the overall hierarchy of controls, with fall soil surface water content being the most important of these. By comparison, when snow cover was low, fall soil surface content was relatively unimportant and superseded by four other controls. Existing empirical methods for predicting infiltration into frozen ground failed to adequately predict runoff response at our site. Our analysis of the hierarchy of controls on meltwater runoff will aid





in focusing new model approaches and understanding what to focus future measurement campaigns on in snowmelt-dominated, seasonally-frozen regions.

# 1 Introduction

Understanding the hierarchies and the time-varying relative importance of controls (*e.g.* topography, vegetation, land use, soil characteristics, and precipitation dynamics) on runoff response is a major challenge in hydrology (Jencso and McGlynn, 2011). Formulating a hierarchy of controls for runoff is necessary for model development (Uchida et al., 2005), a key component of spatial extrapolation (Cammeraat, 2002), and a necessary building block for runoff classification schemes (Barthold and Woods, 2015).

On the Canadian Prairies, spring snowmelt is the dominant runoff-producing event of the year, driving typically 80 % or more of annual runoff (Granger et al., 1984). While some summer runoff is generated by intense, convective rain storms where high rainfall intensities drive infiltration-excess overland flow over localized areas (Shook and Pomeroy, 2012), the controls on these types of events are few in number: rainfall intensity, rainfall magnitude, and antecedent soil moisture conditions (Shook and Pomeroy, 2012). By comparison, the hydrologically more important snowmelt events are much more complicated and affected by multiple interacting factors including snow accumulation, distributed melt inputs, seasonally-frozen ground, ice lenses, and variable pre-melt soil moistures, which combine to produce highly nonlinear runoff responses (Fang et al., 2007; DeBeer and Pomeroy, 2010; Ireson et al., 2013). Consequently, understanding and modeling snowmelt-runoff generation remains problematic throughout many areas of North America and northern Eurasia where snowmelt-influenced, seasonally-frozen ground dominates runoff generation. Nevertheless, in these areas, there is a need to understand snowmelt-runoff generation as it is a critical source of water for human activities and aquatic ecosystems, and snowmelt can cause serious flooding.

In western Canada, snowmelt-runoff has been the subject of many experimental and modeling studies aimed at understanding individual controls: the effects of snow accumulation and redistribution (*e.g.* Pomeroy and Gray, 1995; Fang and Pomeroy, 2009), snowmelt processes (*e.g.* Gray and Landine, 1988), land use and land cover effects (*e.g.* Elliot and Efetha, 1999; Van der Kamp et al., 2003), topography (*e.g.* Shaw et al., 2012), and seasonally-frozen soil (*e.g.* Granger et al., 1984; Gray et al., 2001). However, these studies, and our resultant understanding, are based upon mostly short-term experiments and single-season runoff events. Temporally- and spatially-unstable activation of runoff is the product of nonlinearities and interactions between the various process controls that are not observable in short-term field studies. Much longer records are needed to witness these combinations and interactions of process factors. However, such datasets are rare.



Recently, Coles et al. (2016) presented a 52-year dataset of snowmelt-runoff from three adjacent monitored hillslopes in southern Saskatchewan, Canada. That work showed that long-term snowmelt-runoff and spring soil water content have decreased in response to winter snowfall decreases, while rainfall-runoff has shown no response to changes in rainfall regimes (Coles et al., 2016). They attributed this to the seasonal differences in soil infiltrability, indicating that the controls on infiltration are likely to be most important for snowmelt-runoff amount, as others have shown (Fang et al., 2007; Ireson et al., 2013). However, we still do not know about the hierarchies, interactions, and feedbacks between these controls, and any year-to-year differences in their behaviour. Here, we use this same 52-year dataset to explore these aspects, and contribute for the first time new understanding of the hierarchical importance of runoff controls. And, over a multi-decadal time period, if and how such controls on meltwater runoff interact.

We use decision tree learning (De'ath and Fabricius, 2000) as an investigative tool to extract information from the long-term dataset about the hierarchical controls on runoff generation. Decision tree learning (including classification or regression trees) is an established data mining tool in ecological studies (*e.g.* Spear et al., 1994; Rejwan et al., 1999; De'ath and Fabricius, 2000). It has more recently been incorporated into hydrological studies to leverage process understanding from long-term datasets in temperate regions (*e.g.* Iorgulescu and Beven, 2004; Tighe et al. 2012; Scholefield et al., 2013; Galelli and Castelletti, 2013). To our knowledge, no studies have used decision trees to explore snowmelt-runoff generation. Decision tree learning is fast, conceptually simple, data-based, nonlinear, and non-parametric. Importantly, it allows insights into complexities, nonlinearities, equifinalities, interactions, and feedbacks in the data, which are illustrated clearly in resultant tree-like diagrams (Rejwan et al., 1999; Iorgulescu and Beven, 2004; Michaelides et al., 2009). Here we use the decision tree approach to determine the hierarchies of controls on snowmelt-runoff generation in a seasonally-frozen, snowmelt-dominated region, and any interactions and feedbacks between those controls. Specifically, we focus on the following research questions:

    i)    What is the relationship between annual snow input and snowmelt-runoff output over the 52 years of data?

    ii)   What is the hierarchy of controls on snowmelt-runoff amount?

    iii)  Does the hierarchy vary from year to year or remain constant through time?

    iv)  What are the interactions and feedbacks between the hierarchical process controls?

## 2 Study site and dataset

The study site, known as the Swift Current hillslopes, at South Farm of Swift Current Research and Development Centre of Agriculture and Agri-Food Canada, Swift Current, Saskatchewan, Canada (50°15'53" N 107°43'53" W) on the Canadian Prairies is a set of three adjacent agricultural hillslopes between 4.25 and 4.86 ha in size (Figure 1). Coles et al. (2016) provided




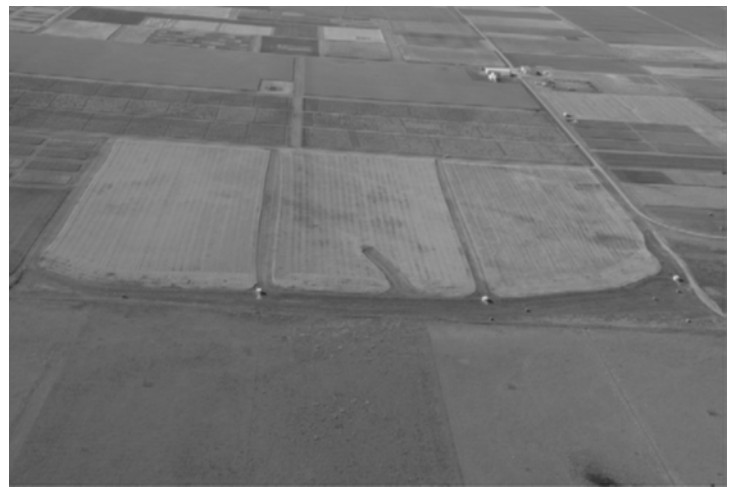

**Figure 1**. Aerial photograph (facing south) of the Swift Current hillslopes (from right to left: Hillslope 1, Hillslope 2, Hillslope 3), taken in a year when wheat was grown. The three small heated huts at the northwest corners of the hillslopes, which house the runoff-monitoring equipment, are visible. Photograph reproduced, with permission, from Cessna *et al.* (2013).

a brief description of the study site, which has undulating topography and shallow north-facing slopes with gradients of 1-4 %. Grassed berms around the perimeters of all three hillslopes prevent runoff from moving between the hillslopes or entering from adjacent land. The only outlet from each hillslope is through a 0.61 m H-flume at the northwest corner of each hillslope. The soil is a Swinton silt loam and classified as an Orthic Brown Chernozem (Cessna *et al*., 2013). The hillslopes typically are

under an annual rotation of wheat (*Triticum aestivum* L.) and fallow. Exceptions to this are: a period (1977-1980) of grass (*Psathyrostachys juncea* (Fisch.) Nevski) and a period (1982-1985) of annual wheat on Hillslopes 1 and 2; an annual rotation (1994-2010) of wheat and legume green manure (*Lathyrus sativus* L.) on Hillslope 1; and an annual rotation (2004-2011) of wheat and pulses (lentils and peas; *Lens culinaris* L. and *Pisum sativum* L., respectively) on Hillslope 2. Hillslope 3 is the only hillslope that has a consistent two-crop rotation and consistent tillage management throughout the 52 years. The hillslopes

have largely been under conventional tillage practice, with the exception of the period 1993-2011 when Hillslope 2 was switched to zero tillage practice. During the period 1993-2004 on Hillslope 2, when the wheat-fallow rotation coincided with the zero tillage period, there was constant standing stubble or standing crop.

From 1962-2013, runoff, snow cover, and soil water content were monitored on the hillslopes. This rich dataset is coupled

with long-term meteorological data recorded at a nearby (*c.* 700 m to the south-southeast) Environment and Climate Change Canada standard meteorological station. Data have been used primarily for studies on the effects of agricultural land management practices on runoff water quality, chemical transport, and soil erodibility (Nicholaichuk and Read, 1978;





McConkey et al., 1997; Cessna et al., 2013). More recently, data were used to study the effects of changing precipitation form and amounts on rainfall- and snowmelt-runoff generation (Coles et al., 2016). Over the 52 years of record, runoff in 22 years was generated exclusively during snowmelt on all three hillslopes (*i.e.* no rainfall-driven contribution to annual runoff on any hillslope), and runoff in 27 years was generated by both snowmelt and rainfall on one or more hillslopes (with an average of

75 % annual runoff from snowmelt). The long-term mean annual snowmelt-runoff depth is 29 mm (Coles et al., 2016). Snowmelt-runoff at this site, and on the Prairies as a whole, is generated as infiltration-excess overland flow when a rapid release of water from the snow cover (usually in a short, one to three week long snowmelt season) occurs over frozen ground of limited infiltration capacity (Granger et al., 1984; Coles et al., 2016).

## 2.1 Meteorological data

Daily (1962-1995) and hourly (1995-present) meteorological data are available from the Environment and Climate Change Canada meteorological station. The data used here include: precipitation (measured using a Belfort weighing gauge), air temperature (daily maximum, minimum, and mean measured inside a Stevenson Screen, and then hourly data measured using a Campbell Scientific HMP35C Temperature and Relative Humidity Probe), wind speed (measured at 2 m and 10 m above ground surface using an RM Young Anemometer Model 05103), and soil temperature (measured at 5, 10, 20, 50, 100, 150

and 300 cm depths using 107B Campbell Scientific Temperature Probes).

## 2.2 Runoff data

Runoff was measured from 1962-2013 with a heated H-flume, stilling well, and a Stevens water level chart recorder at the outflow of each hillslope (Figure 1). Rating curves for each flume/hillslope were used to determine runoff depths (mm) from the stilling well water levels. Runoff depths on hourly, daily, and seasonal timescales (mm) were calculated using a rating

curve for the flumes (following Cessna et al., 2013).

## 2.3 Soil water content data

Gravimetric soil water content was measured twice per year from 1971-2013 on each hillslope. In October (prior to freeze-up) and April (following spring snowmelt) each year, gravimetric soil water content was measured at five increments over the soil profile (0-15, 15-30, 30-60, 60-90, and 90-120 cm) on a permanent nine-point grid on each hillslope. Hillslope-averaged soil

water content at each depth was calculated from the point-scale data. Both hillslope-averaged and point-scale data were recorded from 1980-2013, and from 1971-1979 only hillslope-averaged data were recorded. We converted all soil water content data from gravimetric to volumetric ($\theta_v$) using bulk density data for each depth interval (which ranges from 1.22 g cm$^{-3}$ at the soil surface to 1.51g cm$^{-3}$ at a depth of 100 cm).





## 2.4 Snow cover data

Snow cover depth and density were measured, and snow water equivalent calculated (hereafter referred to as $SWE_C$ for snow water equivalent of the snow cover), for each hillslope during manual snow surveys each year from 1965-2013 on the same nine-point grid as that used for soil water measurements. The means of the nine points were calculated to give three hillslope averages. These were repeated several times from January to March. Measurements from the most recent snow survey before snowmelt were used to calculate the $SWE_C$ on each hillslope at the onset of spring snowmelt.

## 2.5 Quality control

The data were checked and corrected for missing or unrealistic data. During the snowmelt seasons of 1982 and 1985, researchers observed high volumes of snowmelt overwhelming the raised berms causing flow onto Hillslope 2 and Hillslopes 1 and 2, respectively, from adjacent land to the south. For these three occurrences, the total seasonal runoff depth from the hillslope exceeded the depth of total winter snowfall (*i.e.* the runoff ratio exceeded 1). These runoff data were omitted from our analysis and instead given a missing data notation.

## 3 Methods

Decision trees determine a set of 'if-then' conditions between the response and predictor variables and split the dataset according to the largest deviance produced (Rejwan et al., 1999; Michaelides et al., 2009). The result is illustrated in a simple tree-like diagram, with branches, nodes, and leaves. Each final partition (branch) is associated with a certain set of conditions. Branches are composed of nodes. At each node, the dataset is split according to agreement with a single rule (*e.g.* total seasonal precipitation > 50 mm). Splitting continues until the dataset is divided as much as possible. Branches end in a terminal node (leaf), which represents the final partitioning of the data, and the predicted response given the set of conditions dictated by the nodes of the branch. Finally, the tree can be pruned, which removes leaves and nodes with little predictive power, reduces overfitting, and therefore improves predictive accuracy.

To construct the decision trees, we used the 'classregtree' CART algorithm of MATLAB (MathWorks, Inc.). We calculated the runoff ratio, defined as total runoff divided by the total snow water equivalent of the seasonal snowfall (hereafter referred to as $SWE_F$) measured from the start of the hydrological year to the end of snowmelt-runoff, for each hillslope and for each snowmelt season. This resulted in 140 runoff ratios ranging from 0 to 1. We classified the runoff ratios into five equally-sized classes, separated at the 20[th], 40[th], 60[th] and 80[th] percentiles of the runoff ratios. These runoff ratio classes then formed the response (Y) variables for the CART function (Table 1). We used the classification method of CART, so that the predicted outcome is the class to which the data belongs (one of the five runoff ratio classes), instead of the regression method, which is



**Table 1.** Response (Y) variable classes, dependent on runoff ratio ($R_R$).

| Response (Y) variable (runoff ratio class) | Runoff ratio ($R_R$) |
|---|---|
| 1 | $0 < R_R \leq 0.032$ |
| 2 | $0.032 < R_R \leq 0.12$ |
| 3 | $0.12 < R_R \leq 0.28$ |
| 4 | $0.28 < R_R \leq 0.48$ |
| 5 | $0.46 < R_R \leq 1$ |

when the predicted outcome is considered to be a real number. The predictor (X) variables for each hillslope and each season were derived from the long-term dataset (Table 2). An advantageous feature of decision tree construction is that variables can consist of both numerical and categorical data (*e.g.* "fallow" or "wheat" crop types). After the algorithm had divided the dataset as much as possible, we then pruned the tree to a tree size that maximised predictive accuracy and ensured that all leaves were

left with response variable datasets of size N>1. The runoff ratio class at each leaf was the mode of the classes predicted by that branch of the tree.

We first constructed one decision tree (the 'primary' decision tree) using the entire dataset. The controls on runoff ratio were the variables that the decision tree used in its construction. In a second round of decision tree construction, we then took each

of those variables and used them to divide the dataset into two evenly sized halves of the dataset, split by the median of that variable. We constructed two decision trees using these two halves of the dataset. This was to explore directly the reasons for why high or low runoff and runoff ratios might be found under opposite conditions (*e.g.* under both high snowfall and low snowfall conditions). It was also useful for determining the condition-dependent hierarchy of controls.

We quantified the predictive accuracy (or amount of variance successfully explained) of the trees at each leaf using the resubstitution method (Spear et al., 1994). For this, we calculated the percentage of the runoff ratio class predicted correctly at that leaf. The overall predictive accuracy (%) of the tree was the mean of the predictive accuracies at each leaf. While cross-validation is generally the preferred method of estimating accuracy, our dataset was too small to use this method. We quantified the hierarchy of controls by ranking the controls' positions in each decision tree and weighting that rank by the number of

nodes in the tree. If a variable appeared more than once in the tree, we summed the ranking position of each of the nodes at which it occurred, prior to weighting.



**Table 2**. Predictor (X) variables, their descriptions including how they were derived from the long-term dataset, and their minimum, mean, and maximum values.

| Predictor (X) variable name | Units | Description | Mean (Min-Max) |
|---|---|---|---|
| Surface depression storage | mm | Calculated using a 2-meter resolution digital elevation model (DEM) of each hillslope. Hillslope 1 = 2.60 mm; Hillslope 2 = 0.70 mm; Hillslope 3 = 1 mm | 1.43 (0.70-2.60) |
| Topographic wetness index (TWI) | - | Mean hillslope TWI calculated using a 2-meter DEM of each hillslope, following Beven and Kirkby (1979). Hillslope 1 = 5.54; Hillslope 2 = 5.70; Hillslope 3 = 6.03 | 5.76 (5.54-6.03) |
| Land cover | - | Classified as fallow and grass (1) or wheat (2) for the previous summer's crop. | 1.49 (1-2) |
| Fall soil surface water content | (fraction) | Mean volumetric hillslope soil water content ($\theta_v$) in October at the surface (0-15 cm). | 0.154 (0.0713-0.222) |
| Fall soil profile water content | (fraction) | Mean volumetric hillslope soil water content ($\theta_v$) in October for the soil profile (0-120 cm). | 0.123 (0.069- 0.178) |
| Total seasonal snowfall | mm $SWE_F$ | Total snowfall depth from Oct 1st to the end of the runoff period in the following spring, measured at the meteorological station. | 80.5 (37.0-152) |
| Number of warm winter days | - | Number of days each year between Oct 1st and the season's last snow survey that had snow cover (at the meteorological station) and mean air temperature > 0 °C. | 4.53 (0-12.0) |
| Mean temperature on warm winter days | °C | Mean air temperature for days between Oct 1st and the season's last snow survey that had snow cover (at the meteorological station) and mean air temperature > 0 °C. | 2.49 (1.05-5.08) |
| Mean daily wind speed above blowing snow threshold | m s$^{-1}$ | Mean daily wind speed for days when mean wind speed > 7.5 m s$^{-1}$ (minimum threshold for blowing snow redistribution of fresh dry snow on northern prairies; Li and Pomeroy, 1997). | 8.85 (8.25-9.59) |
| Snow cover | mm $SWE_C$ | Mean snow cover water equivalent ($SWE_C$) on each hillslope in the last snow survey before the start of the snowmelt season. | 33.1 (0-121) |
| Spring temperature gradient | °C | Temperature gradient over the seven days prior to the date of peak runoff on Hillslope 2. | 1.66 (-0.0655-5.12) |
| Melt season length | days | Number of days after the season's last snow survey that had snow cover (at the meteorological station) and mean temperature > 0 °C. | 3.80 (1-8) |
| Melt rate | mm d$^{-1}$ | Calculated as: total seasonal snowfall / melt season length. | 26.4 (9.45-111) |
| Date of peak runoff | - | The date when maximum runoff depth occurred, for each hillslope. | Mar 14 (Jan 10-Apr 24) |
| Thawed layer depth | cm | The depth between the soil surface and the top of the frozen layer, at 4pm on the date of maximum runoff, determined using soil temperature data at the meteorological station. | 4.62 (0-147) |





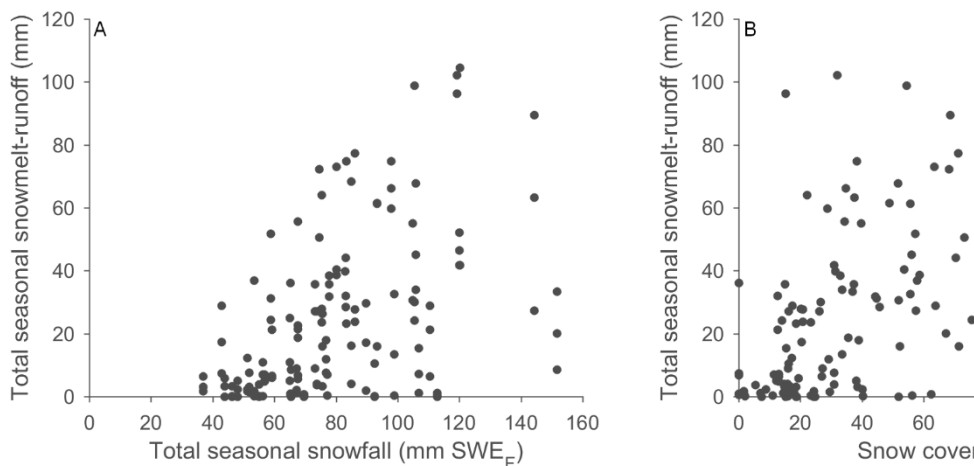

**Figure 2.** Spring snowmelt-runoff events on the Swift Current hillslopes, showing A) the relationship between total seasonal snowfall (mm $SWE_F$) and total seasonal snowmelt-runoff (mm); and B) the relationship between snow cover (mm $SWE_C$) and total seasonal snowmelt-runoff (mm).

## 4 Results

### 4.1 The hierarchy of controls on snowmelt-runoff

Figure 2 shows that there was little relationship between precipitation input and total seasonal runoff output, where inputs were the $SWE_F$ (Figure 2a), and $SWE_C$ (Figure 2b). Six predictor variables were identified by the primary decision tree (Figure 3, Table 3) to explain runoff response: total snowfall ($SWE_F$), snow cover ($SWE_C$), fall soil surface water content (0-15 cm), melt rate, melt season length, and fall soil profile water content (0-120 cm) (Table 4). This resulted in 13 constructed decision trees in total (the primary decision tree using the entire dataset, and six pairs of smaller decision trees). The primary decision tree (Figure 3; Table 3) explained 70 % of the variance of the runoff ratio classes, and contained 10 nodes with six predictor variables and 11 leaves. The 12 secondary decision trees, constructed by splitting the dataset at the medians of each of those variables, identified fewer predictor variables and had fewer nodes and leaves. Their identified controls and hierarchies are given in Table 5. Eight of those decision trees explained more of the variance of the runoff ratio classes than the primary decision tree (Table 6). The decision trees using datasets characterised by high total snowfall, high snow cover, high soil surface water content, and high soil profile water content all were better at predicting high runoff ratios (class 4-5) than their low counterparts, and vice versa for low runoff ratios (class 1-2).





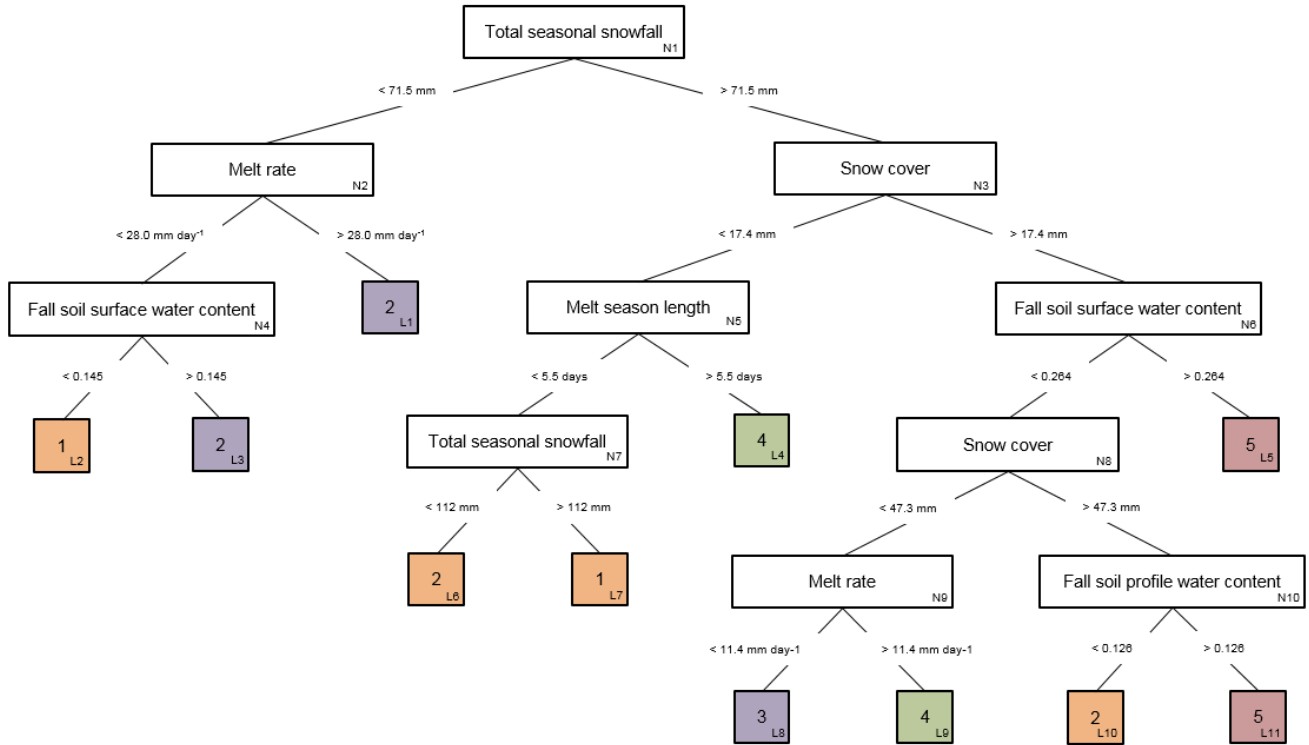

**Figure 3**. Primary decision tree for predicting runoff ratios. The tree shows the variables used to explain runoff ratio, located at the nodes (N1, N2, etc.; white boxes) and the resultant predicted runoff ratio class at the leaves at the ends of each branch (L1, L2, etc.; coloured boxes). Table 3 gives the runoff ratio class data at each node and leaf, and the predictive accuracy at each leaf. The leaf colours are in reference to Figure 4.

Of the (15) predictor variables in Table 2, 12 appeared at least once in any of the decision trees. Snow cover, total snowfall, and fall soil surface water content appeared in the most trees (10, eight, and eight trees, respectively), often with multiple occurrences in any one tree. This confirms that they were important controls on snowmelt-runoff ratio. Surface depression

5      storage, date of peak runoff, and mean daily wind speed appeared in four trees each. Melt rate, melt season length, and fall soil profile water content appeared in three trees each. Lastly, TWI, spring temperature gradient, and land cover appeared in one tree each.

We performed a second iteration of the decision tree construction during which we removed snow cover $SWE_C$ as a possible

10     predictor variable. We hypothesized that this would force the inclusion of any variables that influenced the loss or accumulation of $SWE_C$ on the hillslopes over winter (*i.e.* those variables that controlled the transformation of total snowfall to snow cover). Two variables that were neglected in the original decision trees then became important: number of warm winter days, and





**Table 3.** Runoff ratio ($R_R$) class data at each node and leaf (bold), and the predictive accuracy at each leaf, for the primary decision tree.

| Node number | Parent node | Sample size | | | | | | Predicted $R_R$ class | Predictive accuracy (%) |
|---|---|---|---|---|---|---|---|---|---|
| | | $R_R$ class 1 | $R_R$ class 2 | $R_R$ class 3 | $R_R$ class 4 | $R_R$ class 5 | Total | | |
| N1 | - | 28 | 28 | 28 | 28 | 28 | 140 | - | - |
| N2 | N1 | 18 | 18 | 12 | 5 | 4 | 57 | - | - |
| N3 | N1 | 10 | 10 | 16 | 23 | 24 | 83 | - | - |
| N4 | N2 | 11 | 6 | 2 | 0 | 2 | 21 | - | - |
| **L1** | N2 | 0 | 6 | 4 | 1 | 1 | 12 | 2 | 50 |
| N5 | N3 | 6 | 6 | 4 | 2 | 2 | 20 | - | - |
| N6 | N3 | 1 | 3 | 11 | 17 | 21 | 53 | - | - |
| **L2** | N4 | 9 | 3 | 1 | 0 | 0 | 13 | 1 | 69 |
| **L3** | N4 | 0 | 3 | 1 | 0 | 1 | 5 | 2 | 60 |
| N7 | N5 | 6 | 6 | 3 | 0 | 0 | 15 | - | - |
| **L4** | N5 | 0 | 0 | 1 | 2 | 2 | 5 | 4 | 40 |
| N8 | N6 | 0 | 2 | 8 | 15 | 14 | 39 | - | - |
| **L5** | N6 | 0 | 0 | 0 | 0 | 5 | 5 | 5 | 100 |
| **L6** | N7 | 3 | 6 | 3 | 0 | 0 | 12 | 2 | 50 |
| **L7** | N7 | 3 | 0 | 0 | 0 | 0 | 3 | 1 | 100 |
| N9 | N8 | 0 | 0 | 3 | 13 | 2 | 18 | - | - |
| N10 | N8 | 0 | 2 | 5 | 2 | 12 | 21 | - | - |
| **L8** | N9 | 0 | 0 | 2 | 0 | 0 | 2 | 3 | 100 |
| **L9** | N9 | 0 | 0 | 1 | 13 | 2 | 16 | 4 | 81 |
| **L10** | N10 | 0 | 2 | 1 | 1 | 0 | 4 | 2 | 50 |
| **L11** | N10 | 0 | 0 | 4 | 1 | 12 | 17 | 5 | 71 |

mean temperature on warm winter days. One variable, thawed layer depth, did not appear in any version of the decision trees, which suggested that it was not an important control on snowmelt-runoff ratio.

## 4.2 Condition-dependent hierarchy of controls

The primary decision tree showed that the overall hierarchy of controls was (in descending order of importance): total snowfall, snow cover, fall soil surface water content, melt rate, melt season length, and fall soil profile water content (Table 4). The secondary decision trees showed that the selection of controls and their hierarchy vary when the dataset is split into high or low expressions of those six key variables (Table 5).





**Table 4.** Hierarchy of controls (ranked 1-6) on snowmelt-runoff generation, for the primary decision tree.

| Hierarchy | Control |
|:---:|:---|
| 1 | Total seasonal snowfall |
| 2 | Snow cover |
| 3 | Fall soil surface water content |
| 4 | Melt rate |
| 5 | Melt season length |
| 6 | Fall soil profile water content |

For high and low snow cover years, the hierarchies of controls differed significantly from one another. When snow cover was low, the controls on runoff ratios were largely spring-related (melt season length, date of peak runoff, melt rate, and spring temperature gradient). By comparison, when snow cover was high, the top three controls on runoff ratio matched the overall hierarchy of controls, albeit with differing orders of importance. Further, under high snow cover conditions, fall soil surface water content played the most important role in controlling runoff ratios; whereas under low snow cover conditions, fall soil surface water content was superseded by four other variables in controlling runoff ratios. The hierarchies of controls for high and low instances of total snowfall were relatively similar and retained three of the original six controls from the primary decision tree. They both had snow cover and fall soil surface water content as the top controls, and introduced surface depression storage as a main control.

For instances of low fall soil surface water content, the controls on runoff ratio were quite dissimilar from those in the overall hierarchy of controls. Snow cover, usually an important control, was not important here. Meanwhile, mean daily wind speed did exert a large influence on the prediction of runoff ratios. Further, neither total snowfall nor snow cover controlled runoff ratios under instances of low fall soil profile water content. This indicated that when soil water content was low throughout the soil profile, runoff ratio was not at all predictable based on precipitation amounts. On the other hand, the controls on runoff ratio for instances of high fall soil surface water content and high fall soil profile water content were similar to one another and to those in the overall hierarchy of controls: total snowfall and snow cover were the top two. This indicated that when the ground was wetter than average at the surface and/or throughout the entire soil profile, then the runoff ratio was controlled by precipitation inputs.



**Table 5.** Condition-dependent hierarchy of controls (ranked 1-6) on snowmelt-runoff generation, for all secondary decision trees.

| Total seasonal snowfall | | Fall soil surface water content | | Melt rate | |
|---|---|---|---|---|---|
| High | Low | High | Low | High | Low |
| 1. Snow cover<br>2. Fall soil surface water content<br>3. Melt rate<br>4. Surface depression storage | 1. Snow cover<br>2. Fall soil surface water content<br>3. Total seasonal snowfall<br>4. Date of peak runoff & Surface depression storage | 1. Snow cover<br>2. Total seasonal snowfall<br>3. Fall soil surface water content | 1. Total seasonal snowfall<br>2. Mean daily wind speed & Date of peak runoff<br>4. Surface depression storage | 1. Total seasonal snowfall<br>2. Snow cover | 1. Melt season length<br>2. Mean daily wind speed<br>3. Fall soil surface water content<br>4. Total seasonal snowfall |
| Snow cover | | Fall soil profile water content | | Melt season length | |
| High | Low | High | Low | High | Low |
| 1. Fall soil surface water content<br>2. Total seasonal snowfall & Snow cover<br>4. TWI | 1. Melt season length<br>2. Date of peak runoff<br>3. Snow cover<br>4. Melt rate<br>5. Fall soil surface water content<br>6. Spring temperature gradient | 1. Snow cover<br>2. Total seasonal snowfall<br>3. Land cover | 1. Mean daily wind speed<br>2. Fall soil profile water content | 1. Fall soil profile water content<br>2. Fall soil surface water content<br>3. Snow cover<br>4. Surface depression storage<br>5. Date of peak runoff | 1. Snow cover<br>2. Mean daily wind speed |

When only high melt rate events were analysed, the identified controls on runoff ratio were both precipitation-related (total snowfall and snow cover). Even if a high melt rate was observed, low runoff ratios were still possible if total snowfall or snow cover were low. The highest runoff ratios occurred when there was a high snow cover and a high melt rate. By comparison, the predicted runoff ratios under instances of low melt rates were determined by a combination of different controls, with total snowfall at the bottom of the hierarchy. Even given low melt rates, high runoff ratios could still occur when there was a long melt period and high fall soil surface water content.

When we analysed those years which had short melt seasons, the runoff ratios were strongly controlled by snow cover. A short melt season coupled with high snow cover produced the highest runoff ratio. Otherwise, runoff ratio appears to have been controlled by mid-winter mean wind speeds. By comparison, the runoff ratios in those years with long melt seasons were





**Table 6.** Predictive accuracies for each runoff ratio ($R_R$) class and for the overall tree, for the primary decision tree and each of the 12 secondary decision trees.

| Dataset type | Predictive accuracies (%) | | | | | |
|---|---|---|---|---|---|---|
| | $R_R$ class 1 | $R_R$ class 2 | $R_R$ class 3 | $R_R$ class 4 | $R_R$ class 5 | Overall |
| Primary | 84.6 | 52.5 | 100 | 60.6 | 85.3 | 70.1 |
| High total seasonal snowfall | 60.0 | - | 63.0 | 82.0 | 95.0 | 77.0 |
| Low total seasonal snowfall | 66.7 | 75.0 | 56.3 | 66.7 | 100 | 70.3 |
| High snow cover | 83.3 | 75.0 | 80.0 | 87.5 | 90.6 | 83.6 |
| Low snow cover | 67.5 | 83.3 | 62.5 | 65.0 | - | 72.2 |
| High fall soil surface water content | 100 | 50.0 | - | 60.0 | 91.7 | 78.7 |
| Low fall soil surface water content | 72.2 | - | 50.0 | 57.1 | 62.5 | 62.8 |
| High fall soil profile water content | 100 | 52.6 | - | 63.6 | 70.8 | 71.6 |
| Low fall soil profile water content | 77.8 | 75.0 | 100 | - | - | 82.6 |
| High melt rate | 40.0 | 45.8 | - | - | 50.0 | 45.3 |
| Low melt rate | 100 | 58.3 | 50.0 | - | 69.2 | 67.2 |
| High melt season length | 100 | 66.7 | 50.0 | 58.3 | 87.5 | 70.1 |
| Low melt season length | 66.7 | 46.2 | - | - | 71.4 | 61.4 |

controlled predominantly by soil water content (both profile and surface), followed by snow cover, surface depression storage and finally date of peak runoff. A long melt season coupled with either low soil water content or little snow cover typically led to very low runoff ratios.

### 4.3 Interactions between controls on runoff response

5 We have so far identified the controls, their hierarchy, and how that hierarchy varied under different conditions. This section outlines interactions between controls, *i.e.* whether one variable offset or promoted another variable in the determination of runoff response. While high fall soil water content was associated typically with high runoff ratios, and low soil water content was associated typically with low runoff ratios, these were sometimes mediated by other controls. For example, low runoff ratios could still occur with high fall soil water content if there was little snow cover on a hillslope with high surface depression

10 storage. For winter-time variables, the amount of snowfall and snow cover both had clear effects on the runoff ratio result: typically, high amounts of $SWE_F$ or $SWE_C$ resulted in high runoff ratios, and low amounts of $SWE_F$ or $SWE_C$ resulted in low runoff ratios. However, these were mediated by other controls such that, in eight years, very low runoff ratios (runoff ratio class 1 and 2) resulted despite high snow cover. Finally, for spring-time variables, high runoff ratios occurred when melt rates




were fast, and when the melt period was prolonged and late in the spring. This was especially apparent for high snowfall amounts in the primary decision tree (Figure 3).

For all decision trees, the lowest runoff ratios (class 1) occurred typically when there was either little snowfall or little snow cover on the hillslopes. In some rare occasions, these factors alone triggered low runoff ratios, despite competition from opposing factors that would typically promote high runoff ratios (*e.g.* high soil water content or high melt rate). In other instances, for low seasonal snowfall or snow cover to trigger low runoff ratios, they had to be associated with one or more factors that would also limit runoff ratios. These factors were: a) low fall soil surface water content; b) slow melt rates; or c) high surface depression storage. A very dry fall soil surface water content ($< 0.11$) was always associated with the lowest runoff ratio, regardless of other conditions.

The highest runoff ratios (class 5) always occurred when there was high snowfall or large snow covers. Typically, wet antecedent soil surface water content was also required. If soil surface water contents were low, the highest runoff ratios could still be generated when high snowfall and large snow covers occurred either: on a hillslope with little surface depression storage; or if the entire soil profile (not just the surface) was wet; or if the peak runoff occurred late in the season. For years when winter snowfall amounts were less than the long-term average, the highest runoff ratios could still be generated if soil surface water contents were not low, if peak runoff occurred late (at the end of March or into April), and, finally, if a large proportion of that $SWE_F$ was not retained in the snow cover during the winter (*i.e.* if a large proportion of $SWE_F$ was ablated over winter). This was echoed in a second iteration of the decision tree construction when snow cover was removed from the response variables, in which a condition for the highest runoff ratio to occur was when there were many mid-winter warm days. This indicated that the highest runoff ratios could only occur in low snowfall years when there were mid-winter melts that perhaps raised the soil surface water content or created an ice lens at the soil surface, thus reducing the infiltrability of the soil come spring melt.

Finally, the removal of snow cover from the possible response variables also triggered the inclusion of the land cover variable as an important predictor of runoff ratios; in that case, the highest runoff ratio occurred if the hillslopes were in fallow in the previous growing season. We also analyzed the period of continuous standing stubble on Hillslope 2 (1993-2004) separate from the remainder of the dataset (not, however, using the decision tree approach due to there being only 12 data points of continuous standing stubble). The proportion of the season's snowfall that was retained on the hillslope as snow cover ($SWE_C$ / $SWE_F$) was significantly greater ($p < 0.01$) during the period of standing stubble on fallow (on average, a 0.65 retention) than not (on average, a 0.40 retention), likely due to the snow-trapping qualities of standing stubble. While the period of continuous standing stubble also was associated with, on average, wetter fall soil surface and soil profile water content and higher runoff



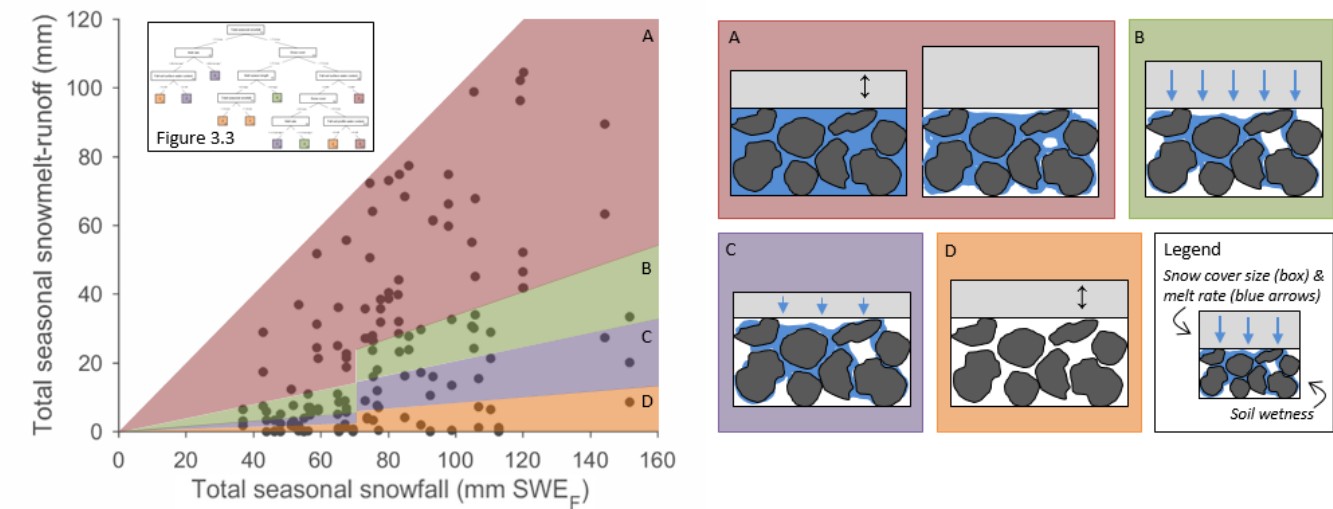

**Figure 4.** Partitioning of the relationship between total seasonal snowfall (mm SWE$_F$) and total seasonal snowmelt-runoff (mm) (Figure 2a), showing typical characteristics of the snowmelt-runoff conditions in different parts of the plot. Colour-coded using leaf colours in Figure 3.

ratios (0.391 compared to 0.254 for the non-standing stubble on fallow instances) there was no significant ($p > 0.05$) difference between the two groups of data.

## 5 Discussion

5  Our 52-year analysis of runoff, precipitation, soil water content, snow cover, and meteorological data showed little relationship between precipitation input and total seasonal runoff output. This highlights the extreme nonlinear relationship between precipitation inputs and runoff outputs on frozen ground at the hillslope scale. The additional factors that controlled runoff partitioning included total snowfall, snow cover, fall soil water content at the surface and through the soil profile, melt rate, and melt season length. Together, these explained 70 % of the runoff ratio variance, and accounted for the scatter in the

10  precipitation-runoff relationship (Figure 4). Further, these factors were hierarchical and condition-dependent. For example, in years when snow cover water equivalent (SWE$_C$) was high, fall soil surface water content played the most important role in controlling runoff ratio. In years when snow cover was low, fall soil surface water content was relatively unimportant and was trumped by four other variables.



## 5.1 Infiltration into frozen soil controls hillslope runoff ratio

Three groups of variables controlled snowmelt-runoff ratio: precipitation amount (represented by total snowfall and snow cover), antecedent wetness condition (represented by fall soil surface water content and fall soil profile water content), and melt intensity (represented by melt rate and melt season length). Together, they determined collectively the balance between the fraction of precipitated water that infiltrated and the fraction that ran off and was delivered to the hillslope outlet. These six key variables controlled infiltration. While others have shown infiltration to be a key factor in hydrological partitioning of snowmelt water (Granger et al., 1984; Fang et al., 2007; Ireson et al., 2013), our long-term analysis is the first to show the hierarchical importance of controls in controlling infiltration.

Infiltration capacity is known to change through an event (Horton, 1933). During a water input event where soil is saturated from above, infiltration capacity typically is high at the beginning of the event, followed by a rapid decline and asymptotic reduction (over minutes, hours, or days) to a near-constant value and quasi-steady-state (Zhao and Gray, 1997; Dingman, 2008). This steady state value is the saturated hydraulic conductivity. The decline in infiltration capacity is due to sorptivity – the potential of the soil to absorb and transmit water through capillarity. This is higher for dry than for wet soil. For water input to frozen soil, an additional factor linked to infiltration capacity decline is the re-freezing of meltwater in soil pores causing blockages (Ireson et al., 2013). Nevertheless, the shape of frozen soil infiltration curves is similar to that of frozen soil (Kane and Stein, 1983).

Figure 5 shows a conceptual model of how each control in turn influences runoff ratio via the process of infiltration, where a constant melt rate that exceeds the infiltration capacity of the soil is assumed. The greater the precipitation event amount (where the event is the melt season and the precipitation amount is the depth of snow cover or the total seasonal snowfall) or the longer the melt season, the more of the declining infiltration capacity curve is traversed with time (along the x-axis). Figure 5b shows a scenario where a large melt event generates greater amounts of runoff, and greater runoff ratios because of the relative volume of meltwater that infiltrates (the integral of the infiltration curve) versus that which does not. Similarly, melt rate determines how much precipitation, at any point in time, exceeds the infiltration rate of the soil (y-axis of the curve). Figure 5c shows that a higher melt intensity exceeds the infiltration rate for a longer period of time, and produces greater runoff ratios. Finally, antecedent soil water content (and other soil characteristics) control the shape of the infiltration rate curve. Figure 5d shows that wet antecedent soil conditions cause reduced initial infiltration rates, which are more readily exceeded by melt rates and thus lead to higher runoff ratios. This simple conceptual model of infiltration explains why these six variables – snowfall, snow cover, melt season length, melt rate, fall soil water content at the surface and through the soil profile – exert the greatest control on runoff ratio.





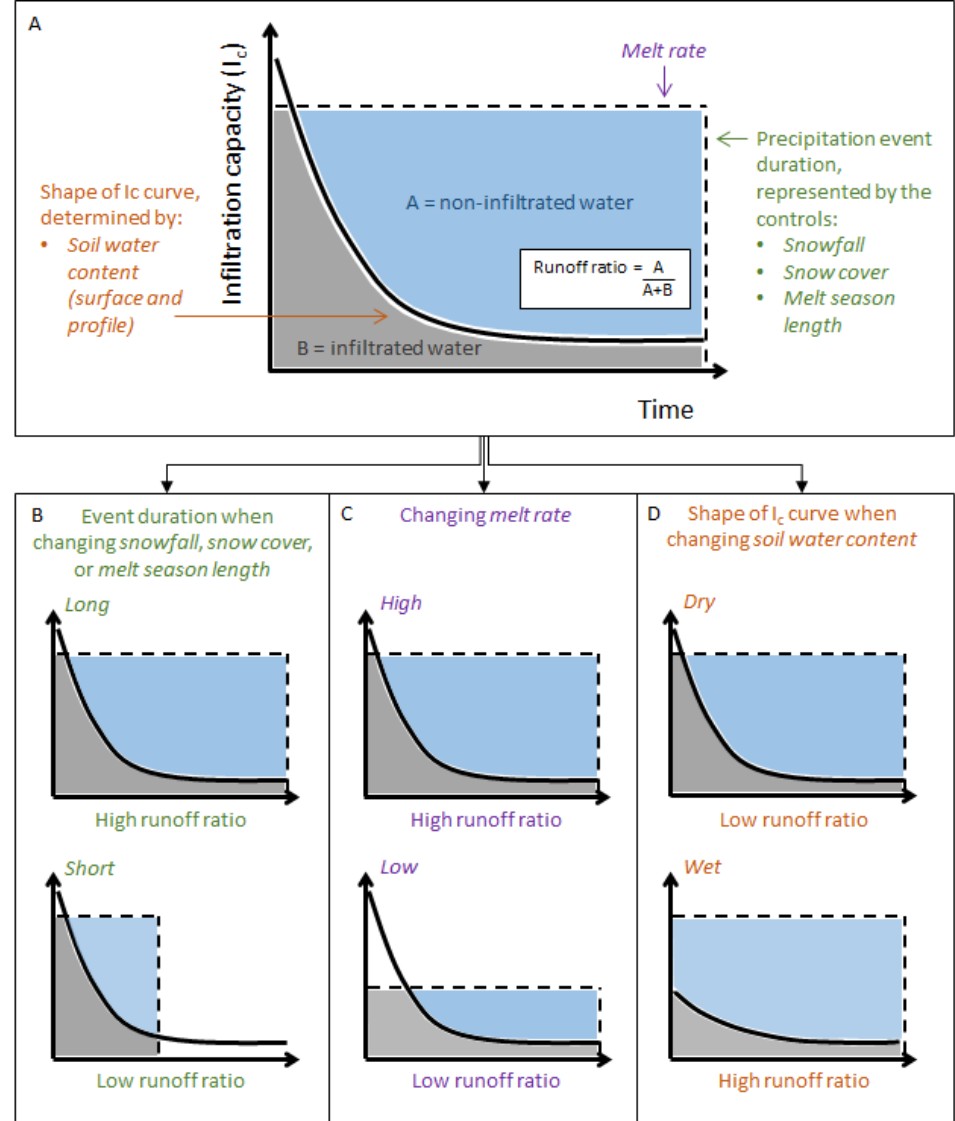

**Figure 5.** Conceptual figure showing how the key controls on runoff ratio affect runoff ratio via infiltration. A) Features of a typical infiltration capacity curve; B) snowfall, snow cover, and melt season length control event duration (green), with all other factors being equal; C) melt rate (purple) controls the incoming water flux; and D) soil moisture affects the shape of the infiltration curve (orange). When just one of these is varied (and the other are held constant), the volume of water that infiltrates (brown) and the volume that is in infiltration-excess (blue) changes, and it is the balance between these two that dictates the runoff ratio.

Varying snow cover and antecedent soil moisture conditions seemed particularly influential in causing shifts in the hierarchical ordering of controls. Under high snow cover conditions, fall soil surface water content played the most important role in



controlling runoff ratios. By comparison, under low snow cover conditions, fall soil surface water content was superseded as a control on runoff ratio by four other, largely spring seasonally-related, variables. The infiltration rate curve helps explain why this might be the case (Figure 5). Given a decline in infiltration over time, for a small amount of snow cover, the infiltration curve is only traversed at the start (where infiltration rate changes quickly over time), so any change in melt rate exerts a large effect on the resultant runoff ratio. This is compared to a large amount of snow cover, where runoff ratios are less sensitive to changes in melt rate. This would explain why melt rate was identified as being a stronger control on runoff ratios under conditions of lower snow cover.

Our results showed that when fall soil water contents were low, runoff ratios were not predictable based upon the usual controls on runoff ratio. For example, runoff ratios when the entire soil profile was dry were not predictable based on precipitation amounts. Runoff ratios when the soil surface was dry were not predictable based on snow cover or the actual soil surface water content. This further indicates that infiltration into frozen soil, already a difficult flux to predict, was especially variable when the soil was dry in the fall. Further, while soil water content was the most important control on runoff ratios under conditions of high snow cover, it was a relatively unimportant control under conditions of low snow cover. When snow cover at the end of winter was low this could imply either little seasonal snowfall or significant ablation over the winter. Ablation might have caused over-winter changes in soil water content, thus rendering the measurement of the fall soil water content an imprecise or misleading representation of the pre-melt water content. This would in turn lead to its reduced importance in the decision tree. The snow cover water equivalent on the hillslopes was typically significantly less than the total seasonal snowfall water equivalent, illustrating the importance of mid-winter ablation events (*e.g.* by melt, sublimation, or snow redistribution by wind) and supporting the findings of Gray et al. (1983; 2001) and Pomeroy et al. (2007a).

## 5.2 Soil moisture memory

Notwithstanding these likely mid-winter changes to soil water content, soil moisture memory from fall freeze-up was important. We observed that fall soil surface water content, measured immediately prior to temperatures falling below freezing and before the onset of snowfall, was the third key control on spring (four to six months later) snowmelt-runoff response. This is consistent with observations from more humid regions, where soil moisture memory has been analysed and described in the context of the persistence in the soil of anomalous wet or dry conditions that have long since been forgotten by the atmosphere (*e.g.* Entin et al., 2000; Mahanama and Koster, 2003; Orth and Seneviratne, 2013). These studies, largely based on modeling approaches or long-term data analysis, have shown that memory in mid-latitude regions is strongest under extreme (particularly extreme dry) conditions (Wu and Dickinson, 2004). Memory timescales have been reported up to 2-3 months (Mahanama and Koster, 2003; Vinnikov et al., 1996; Entin et al., 2000). Also, soil and vegetation characteristics have been shown to be more important than the climate regime in determining the soil moisture memory strength (Orth and Seneviratne, 2013).





That fall soil water content in cold, snow-dominated seasonally- or permanently-frozen ground locations was a key control on spring runoff is not new; a common assumption of hydrological modeling approaches is that the soil water content at the start of snowmelt (typically March-April) equals the soil water content at the time of freeze-up (October-November). In other words, the water content is believed to be 'locked in' through the winter and remain constant until spring thaw, and thus exhibits soil moisture memory. Of course, we know this to not be necessarily true due to the likelihood of mid-winter melt events having caused increases in soil water content.

Further, vapor transfer across a soil-air or soil-snow interface, and mid-winter vertical redistribution of water within the soil towards the downward-advancing freezing front all challenge this assumption (Kane and Stein, 1983; Gray et al., 1985; Quinton and Hayashi, 2008; Nagare et al., 2012). Gray et al. (1985) suggested that in the absence of mid-winter melt events, soil water content in the 0-30 cm surface layer decreases over winter, with the greatest losses (only a 3-4 % decrease, typically) seen for fallow (as opposed to stubble) lands. We hypothesize that this is due to further infiltration to deeper parts of the soil profile. We found that there was a strong positive relationship between the length of time during which the soil was frozen over winter and the runoff ratio in the following spring, with the relationship strongest for 50 cm and 100 cm depths (data not shown). This might indicate that a longer winter period of deep soil freezing prevented the soil surface water content from dissipating vertically, thus retaining the soil moisture at the surface, and driving a high runoff ratio come the spring. It might also indicate that faster infiltration rates and reduced runoff ratios – when the wetting front reached the thawed layer below the frozen layer (Watanabe et al., 2012) – were not reached because the wetting front did not reach thawed soil when the soil was frozen to greater depths. The longer the soil was frozen and the deeper it was frozen, the stronger the soil moisture memory was. If the fall soil water content was high, this then drove higher runoff ratios in the spring.

Despite mid-winter melt events and land cover causing deviations between fall and pre-thaw soil water content, fall soil water content remained an important control on spring runoff ratios. This indicates long soil moisture memory in the system, which was heightened the longer the soil profile was frozen down to 100 cm. This soil moisture memory propagated through to runoff response. However, the soil moisture memory observed here, we believe, ought to be distinguished from existing descriptions of soil moisture memory from elsewhere (*e.g.* Entin et al., 2000; Mahanama and Koster, 2003; Orth and Seneviratne, 2013). This is because those could be thought of as 'active' systems, while our observations are from a more 'dormant', frozen system. For this cold, seasonally-frozen region, soil moisture memory was less about the persistence of an anomaly, and more about dormancy. Hence, memory of the system was due to climatic conditions, more so than the soil and vegetation characteristics.





### 5.3 Implications for modeling and future field campaigns

Our findings have implications for existing approaches for predicting runoff responses to snowmelt events, especially regarding the ways existing models deal with infiltration and regarding what (and when) experimentalists ought to focus their observations on in the field. We do not believe that the nonlinearity and condition-dependent nature of these controls defies our ability to model, since we have shown here that it can largely be explained in the context of infiltration. Several physically-based approaches exist for determining infiltration into frozen soil, including models that solve heat and water transfers through porous media such as GeoStudio (GeoSlope International, 2015), SUTRA (Voss and Provost, 2002), SHAW (Flerchinger and Saxton, 1989), SOIL (Stähli et al., 1999), and HYDRUS 1D (Hansson et al., 2004), and models that use the pore size distributions and other physical aspects of the soil such as capillary bundle models (Watanabe and Flury, 2008). These approaches require considerable data to drive the energy and meteorological inputs, and to model the soil domain; much of which is not available here.

In hydrological studies on the Canadian Prairies, commonly implemented models are empirical equations, such as that of Granger et al. (1984):

$$I = 5(1 - \theta_{va}) \, SWE_C^{0.584} \tag{1}$$

where $\theta_{va}$ is the pre-melt antecedent volumetric soil water content in the 0-30 cm soil layer, and total infiltration ($I$) and snow cover water equivalent ($SWE_C$) are in millimeters. This equation drives the infiltration module of the Cold Regions Hydrological Model (Pomeroy et al., 2007b), a widely implemented model for snow- and snowmelt-dominated regions (*e.g.* in Canada: Ellis et al., 2010; Quinton and Baltzer, 2013; Fang and Pomeroy, 2008; in China: Zhou et al., 2014; in Europe: Lopez-Moreno et al., 2014). If we test this equation with our data to calculate its efficacy for determining the component of the water balance that infiltrates into frozen soil we can rearrange this to calculate runoff ratio ($R_R$) as we defined it in our analyses above, using the total seasonal snowfall ($SWE_F$):

$$R_R = 1 - \frac{5(1 - \theta_{va}) \, SWE_C^{0.584}}{SWE_F} \tag{2}$$

According to Granger et al. (1984), two controls determine snowmelt-driven infiltration and runoff ratio: snow cover and soil water content. Our results support that these two variables are indeed key controls on runoff ratio (in our case, these controls were ranked second and third, respectively, in the hierarchy of controls). However, using Eq. (2) to predict our observed runoff



ratios over the 1971-2013 period explained only 14.3 % of the variance of the runoff ratio classes (data not shown). It overestimated low runoff ratios, and underestimated high runoff ratios.

Therefore, this frequently-used equation for infiltration into frozen soil is of limited use. This is not surprising since, based on our results, the determinants of runoff ratio are more complex. While a more mechanistic model is certainly needed to bring in these elements, doing this in a deterministic way would be the basis for future work. The results of this paper's decision tree learning – the key controls the decision tree has identified, the condition(s) under which each control are important, and the ways in which they interact and feedback between one another – could be a way to frame a model structure for snowmelt-runoff over seasonally-frozen hillslopes.

In terms of what to measure to parameterize new models, we can use our hierarchy of controls to guide cost-effective and useful field measurements. While seemingly obvious, this study has reinforced the need for reliable snowfall and snow cover data. It also has emphasized the critical need for measuring pre-freeze soil water content in the fall: surface observations (0-15 cm) are most important, but total soil profile water content would also be beneficial. The influence that mid-winter melt events appeared to have on soil water content and ice lens creation means that pre-melt soil water content should also be measured, if possible. This measurement is most important when the soil is relatively dry in the fall, when mid-winter melt events occur, or when there is a large amount of snow cover. However, accurate measurement of unfrozen and frozen water content in frozen soils in field conditions remains a problem, with probes (for example, dielectric instruments or gamma probes) requiring significant and complex calibration (Ireson et al., 2013), and manual, sample-extraction methods proving very difficult given the frozen nature of the ground and the overlying snow. With any advances in instrumentation and methods for more reliable measuring of soil water content in frozen ground, these should be deployed to track antecedent moisture conditions through the winter and aid in the prediction of snowmelt-runoff response. While these fall-based and winter-based observations are most important for the prediction of runoff response, the spring conditions are of course also key to the response, especially when there is a small amount of snow cover. For predictive purposes, the melt rate and duration of the melt season need to be estimated in advance. For development, calibration, and/or validation purposes, these need to be documented.

## 6 Conclusions

We examined a 52-year dataset of runoff, precipitation, soil water content, snow cover, and meteorological data to determine the hierarchy of controls on snowmelt-runoff generation. Our decision tree analysis showed that the most important controls on snowmelt-runoff generation were, in descending hierarchical order of importance: total snowfall, snow cover amount, fall





soil surface water content, melt rate, melt season length, and fall soil profile water content. Together, these were able to account for the scatter in the precipitation-runoff relationship. The hierarchy of these controls was controlled by actual conditions, with the biggest hierarchical differences between high and low snow cover seasons, and wet and dry antecedent conditions.

The key variables determining the runoff ratio collectively reflected the controls on the fraction of precipitated water that infiltrated. Despite some mid-winter meltwater likely increasing the soil water content, the system showed significant memory in that the soil water content in the fall was a strong control on runoff in the spring. Here soil moisture memory was mostly determined by system dormancy and less so by the persistence of an anomaly. This distinguishes soil moisture memory in our system from that in more humid regions. An existing commonly-used method of predicting infiltration into frozen soil (Granger

et al., 1984) did not adequately predict runoff ratios in our case, which has implications for its inclusion in hydrological modeling. There is therefore room for new empirical models. Our results showed field-based measurements for estimating snowmelt-runoff response must include pre-freeze soil water content (primarily at the surface but also through the entire soil profile, if possible), snowfall and snow cover water equivalents, pre-melt soil water content to account for any over-winter changes in the soil water content, and, through the spring snowmelt season, melt rate and melt season duration.

**Acknowledgements**

We thank the many Agriculture and Agri-Food Canada researchers, technicians, and students in Swift Current who have collected these data over the last 52 years. In particular, we acknowledge Don Reimer and Marty Peru who were the responsible technicians for 1971-1992 and 1993-2011, respectively. We thank Markus Weiler for comments on an earlier draft of this paper. The data analysis work was supported by an NSERC Discovery Grant awarded to JJM.

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
