# Peer review of "The hierarchy of controls on snowmelt-runoff generation over seasonally-frozen hillslopes"

_Hydrology and Earth System Sciences, 2016_

## Referee Comment (RC1) · Anonymous Referee #1 · 6 Dec 2016

The manuscript aims at identifying the relative importance of controls on snow-melt runoff from three arable plots of approximately 5 ha located in the Canadian Prairies. To this end they use a statistical method – decision trees – with a set of 15 predictor variables to analyze nearly 50 years of runoff data. The authors conclude that the relative importance of controls is not general for all winters, but changes (for example) with the thickness of the snow cover.

As the authors correctly show in the introduction, there has been a substantial amount of experimental studies in the past 30 years on winter-time runoff generation with a focus on the snow cover and the frozen soil – at different scales (from small plots to catchments) and in different regions of the world (incl. Canadian Prairies, Northern Scandinavia, Alpine areas and Japan). So, I dare say that we know quite a lot already about important controls of snow-melt infiltration and runoff – including formation and

permeability of a frozen soil layer. Therefore my first reaction when I started reading this manuscript was: do we really need a statistical analysis to find out the "hierarchy of controls" on snowmelt-runoff for seasonally frozen areas? And, what's the added value of such a statistical analysis to the existing knowledge from empricial studies and deterministic modelling? After having finished reading the manuscript I have the feeling that the lessons learned from this exercise are marginal. This has to do with a number of critical drawbacks of this study:

a) The number of winter runoff situations (response variable) is critically low compared to the large amount of predictors (15).

b) Some of the predictor variables are highly correlated with other predictor variables.

c) We know from many field experiments that runoff in spring is not the result of average winter conditions, but can reflect critical short situations of the winter; e.g. short mid-winter melt events, or short coincidence of a shallow snow pack and very cold air temperatures generating substantial soil frost. This applies in particular to regions with a broad range of soil frost conditions, such as the Canadian Prairies, that are sensitive to the snow-cover thickness. In conclusion, an analysis based on important predictor variables representing mean winter conditions (e.g. mean temperature or total seasonal snowfall) is probably not very conclusive.

There are a few other issues with this study that I consider as critical:

- The title and some parts of the manuscript imply that the relative importance of the different controls on snowmelt-runoff generation – found in this study – are general. But in fact, it only applies to the specific slope, soil and meteorological conditions of this field site. How can the hierarchy of controls found at this site be transferred to other sites with other snow and soil conditions, or with another topography?

- The reader gets very little information about the specific conditions of this site. In fact, it seems that what is called "snow-melt runoff" is only "surface runoff". What about

lateral discharge? Is there a groundwater table fluctuating near the surface? What do we know about the soil type and the soil physical properties? How does a typical snow cover look like in this area? I guess it must be very patchy and loose. What's the typical length of the winter season? Such information would be essential for interpreting the results.

- Two of the most important predictor variables are "Total seasonal snowfall" and "snow cover (i.e. peak SWE)" (see Fig. 3) – (which is by no means a surprise) – and I assume that these two variables are highly correlated. How do the authors justify the selection of these two predictor variables?

- One key-variable for snowmelt-runoff generation at this site is completely missing in the analysis (and also in the discussion!): the extent of the soil frost. And with that I mean both the frost depth and the ice content. I didn't find any information about typical frost depths, for example. Of course, there is an implicit account of soil frost because of the strong (negative) relationship between snow cover and soil frost depth. And there is an implicit relationship between pre-winter soil water content and ice content later in the winter. But this obvious key-connection is not made.

- How about land-use? According to chapter 2 the three plots were covered with different vegetation and experienced different tillage practices. But it seems that this was not of major importance for the snowmelt-runoff generation. (Also the topography of the three slopes is more or less the same.) If that's the case, I think that the decision tree actually has only an N of 52 (and not 140), which makes the statistical analysis even more questionable. However, if vegetation and tillage have a significant impact, then it should be also discussed somewhere.

- Chapter 5.3: Implications for modeling and future field campaigns: are we going to change current practices in modelling snow-melt runoff (in regions with seasonally frozen soil) based on the lessons learned from this statistical analysis? I don't think so. The key-role of the snow cover and the significance of pre-winter soil moisture content

has been known for quite a long time and is accordingly represented in current runoff models. And also the critical need for accurate and spatial snow-cover data is well recognized.

- Finally, the world-wide knowledge on snowmelt-runoff generation in areas with seasonally frozen soil is not well reflected in the introduction. Only Canadian studies are referred to.

In conclusion, I'm not convinced that the above-mentioned problems can be solved with the available data and the selected method to become a contribution that adds value to the existing state-of-the-art knowledge on snowmelt-runoff formation in cold regions.

---

## Short Comment (SC1) · 7 Dec 2016

**Review of the research paper "The hierarchy of controls on snowmelt-runoff generation over seasonally-frozen hillslopes – by Coles et. al."**

**Summary**

Schälchli, S.

Coles et. al. used a 52-year dataset from a research on a farmed, undulating hillslope in northern Canada which contained multiple measurements of meteorological, runoff, soil water content and snow cover parameters (Coles, A. E. et. al. (2016): 3-6). They were interested in all these determining factors, which influence the snowmelt-runoff. Their target was to find out "the hierarchies of controls on snowmelt-runoff generation in seasonally-frozen, snowmelt-dominated region, and any interactions and feedbacks between those controls" (Coles, A. E. et. al. (2016): 3).

For implementing their research questions they used decision trees to identify the hierarchies in the influencing variables. For the construction of the decision tree Coles et. al. used the CART algorithm of MATLAB (Coles, A. E. et. al. (2016): 6). They defined 5 different runoff classes (where as the first was the lowest runoff). By splitting the first decision tree, which contained the whole information, by the median they were able to explore the causes for the low or high runoff ratios under opposite conditions. They defined several determining factors, which were tested on their hierarchical influence on the snowmelt-runoff and under which conditions which factors were dominant (Coles, A. E. et. al. (2016): 7-9).

Coles at. al. found out that in this long term analysis data show little connection between precipitation and total seasonal runoff output (Coles, A. E. et. al. (2016): 9). This indicates the immense nonlinear relationship on seasonal frozen ground between precipitation inputs and runoff outputs (Coles, A. E. et. al. (2016): 16). The main factors which influenced the runoff output were the total snowfall, snow cover, melt rate, fall soil water content trough the whole soil profile and at the surface and as well the melt season length. All this factors together accounted for 70 % of the runoff ratio variance (Coles, A. E. et. al. (2016): 9-10).

Their main finding in their study was that the fall soil surface water content was mainly important in direction to the snowmelt-runoff (as the following figure 1 shows) (Coles, A. E. et. al. (2016): 16).

[Figure]

Coles, A. E. et. al. (2016): 16

They recognised that especially when the snow cover was high the soil surface water content was the mayor control for the snowmelt-runoff in spring. Reverse when we had just small snow cover during the winter season the fall soil surface water content was not so important for the runoff. This is because in this case the runoff output is mainly controlled by spring related controls as melt season length and the peak date of the runoff (Coles, A. E. et. al. (2016): 12-15).

Summarizing we can say that the soil moisture content is very important for the snowmelt-runoff in seasonal frozen hillslopes (Coles, A. E. et. al. (2016): 23). Just because of the high importance that soil moisture content (before the melting season) have on the prediction of runoff outputs, it is particularly important to observe the still existing uncertainties concerning the influence of infiltration rates into the soil during winter. Thus in further studies the scientist should also measure the soil water contents (on surface and in the hole profile) during winter months to determine the influence of ablation events as sublimation, melting or snow redistribution by wind (Coles, A. E. et. al. (2016): 16-20, 23).

**Review**

The article includes many very interesting contents and also adds new results to the broader scientific context. Furthermore they came up with new ideas and used tools, which were never used in this research context before. The Title reflected the contents of the paper clearly.

In the beginning the article was quite hard for me to read. I had to read some paragraphs twice and look up some technical terms. Especially the abstract uses, from my point of view a quite difficult language and the theme often jumps among the different results.

The Introduction was written clearly and the authors made a good assumption of the present scientific stage. Furthermore they linked their research to other papers and declared why their research topic is of relevance. They generated a very good embedding of their topic into the broader scientific range. The research questions were pointed out clearly. But the 3rd Question was answered insufficient, because the author didn't mention in what way the hierarchy varied from year to year.

The used data was in the majority good, but as they also made clear, the measurements of the soil water content were made a bit too seldom. The measurements have been done in October (before the freeze-up) and then only in April when the melting process has already begun in many years. This is not an ideal presupposition for the correctness of the soil water content and therefore for the whole influence of the soil moisture on the runoff output. The reason is that until April there may occurred infiltrations (rising of soil water content) or percolations (decrease of soil water content).

Another question crossed my mind when I read the chapter 2 on the study side and the dataset. Is there maybe a difference in the snow adhesion by the diverse crop types stubbles? For example in term of the stability respective to wind redistribution?

I have to point out that I wasn't completely able to follow up with the methodology part. For example it was unclear to me how the authors came up with the decision in which hierarchy the controls are in an explicit case and I also had my difficulties to read the decision tree. First when I read the whole paper trough and then returned to the abstract or to the decision tree the intentions became clearer.

The results and the conclusions were written very comprehensible and contained many good visualisations and graphics. The Figure 4 and 5 were in particular very illustrative and informative. However why there is a sharp step of the seasonal snowmelt-runoff at 70 mm is unclear to me. Their results were highly interesting, especially in terms of the influence of the soil water content under the seasonal frozen conditions on the runoff output. I support their appeal to measure also the pre melt soil water content (Coles, A. E. et. al. (2016): 23). Than during reading the results I was thinking about the faults which may occur when the soil water content is only measured twice a year and thus was very happy to find my thinking's reinforced in the conclusion part by the authors themselves.

Finally the conclusions were hold shortly and precise with a clear statement addressing further research on the influence of soil water content in seasonal frozen soils on the runoff output.

**Recommendation List (red = major points, green = minor points):**

- Central theme in abstract have to be found and a better organisation of the results (in Abstract) should be done

- 3rd Research question: Variations in between of the different years were not addressed

- Method part should be better clarified and the functioning of the decision tree should be explained better

- Figure 4: Author should explain the occurring sharp step at 70 mm in the manuscript. And are the shapes in the soil boxes the granularity of the soil or ice lenses? Label it clearer.

- Comparison of the different crop types and their effect on snow cover and runoff

- Interesting would be how the determining factors changed over time and if there is a trend in the data (maybe caused by climate change?). If there is a trend: How does the change in the determining factors influence the snowmelt runoff?

**Literature**

Coles, A. E. et. al. (2016). The hierarchy of controls on snowmelt-runoff generation over seasonally-frozen hillslopes. Hydrol. Earth Syst. Sci. Discuss. In Hydrology and Earth System Sciences HESS. pp. 1-27.

---

## Referee Comment (RC2) · Anonymous Referee #2 · 10 Jan 2017

The authors use a decision tree method and a 52-year data record of snowmelt related runoff from three agricultural plots in the Canadian prairies to study the controlling factors on runoff generation over seasonally frozen soils. Spring runoff conditions are characterized by determining runoff ratios. Decision trees are then used to understand the relative importance of a number of control parameters describing topography, land use, vegetation, and precipitation dynamics on the spring runoff. The intent of the study is to improve the understanding of the relative importance of the factors responsible for runoff over frozen grounds which should help to improve models describing infiltration into frozen ground and runoff over frozen hillslopes. The authors also want to give some advice to future measurement campaigns looking at seasonally frozen, snowmelt dominated regions. The authors claim that the major innovative part of their study as compared to other studies looking at runoff generation cover frozen ground

is, that those studies usually are based on short term experiments and single season runoff events, while their study looks at averages over a much longer data period. This longer record should help to reveal the nonlinearities and interactions between the various process controls. In my opinion, this aspect of the study could indeed present some important new insights into the scientific field of snowmelt runoff generation over frozen ground. The paper is generally well written and the language and citing are appropriate. However, I find parts of it (especially in the Methods section) rather difficult to follow (see comments). The introduction and literature review focuses very narrowly on studies mostly from western Canada. While this is understandable due to the study location, I think a broader look at studies of runoff over frozen soils would be beneficial. The research goals on the other hand are clearly stated and well defined. The results and discussion section are adequate but could maybe be structured a little more clearly (see comments). The conclusions are based well on the results and provide a good summary of the study. The Tables are concise and easy to interpret, while some of the Figures could be improved (see comments) to make them easier to grasp. The topic of the article falls within the scope of the journal and does in my opinion represent a worthwhile contribution to the snow science community. However, the article needs in my opinion some fairly substantial changes to make it less difficult to follow. I would therefore recommend publication after major revisions (see general and specific comments).

General Comments

As mentioned, the literature review in the introduction should be broadened to include at least some studies of runoff and runoff generation over seasonally frozen or even permafrost soils. I would also welcome a discussion of the relevance of the studies result for other regions of the globe. Are the study results only applicable to prairie landscapes or could similar hierarchies and interactions be found for example in high arctic locations that have frozen soils and often similar topography. This discussion could also be included towards the end of the paper in the general discussion section.

I found it somewhat hard to follow the explanation of the decision tree method in the Methods section. The description is very technical, which makes it hard for the average reader to understand what the analysis really means. I think it would be beneficial to include some explanations as to what it means f.e. if some predictors occur several times in a tree, what the ranking of the nodes mean for the importance to runoff generation, and what a leaf really represents "in reality". I guess I would recommend to "dumb down" the explanation a little bit so that readers who are not familiar with the decision tree technique can appreciate what the single tree elements (nodes, leafs etc.) mean for the description of the "real life" process. I would really encourage a much more in depth presentation and discussion of the predictor variables as part of the discussion. The choice of which predictor variables to use is a major point of the study and should be treated as such. So far information on the predictors is only available within Table 2 and is very short. I would especially welcome a short description for each predictor, why they were chosen, and how they theoretically are expected to influence runoff. Additional points that could be discussed here are: What processes are believed to lead to the differences between total snowfall SWEf and snow cover SWEc. How closely are these two predictors related (maybe include an autocorrelation analysis). What impact does the soil fall water and fall soil profile water content have on the mechanics of spring infiltration (i.e. formation of concrete ice soils when water content is high, this might not be obvious to readers not that familiar with frozen soil hydrology)? Why is mean daily wind speed needed? The process deposition or removal of snow from the test plots due to blowing snow should be included in the difference between total snowfall SWEf and snow cover SWEc. So what additional information or impact does blowing snow during the winter have on spring runoff. The authors use 4 predictors to describe melt conditions. Wouldn't melt season length and melt rate be enough? What additional information do the other two predictors provide?

Specific Comments

p.4 Line 5 and following The authors describe the usual conditions on the test plots

as they were implemented in the decision tree analysis and the exceptions to those conditions. Unfortunately, considering the study period of 52 years there seem to be quite a lot of extraordinary conditions considering land use and tillage. One would assume that especially different tillage practices and the presence of stubble or standing crop over the winter (additional catch of blowing snow as mentioned later in the paper) could have quite an impact on runoff. Yet it seems like all years were included in the analysis. At the very least I would expect a discussion of the possible impacts of this. p.6 line 25 The runoff ratio is defined as total runoff divided by SWEf from the end of the hydrological year to the end of snowmelt. How was liquid precipitation (i.e. rain) handled in this? I'm aware that in Saskatchewan rain after the end of the hydrological year and before the start of the snowfall and during snowmelt is rather rare, but it must have occurred sometimes during the 52 year study period. Also how was SWEf measured at the met station (precipitation sampler, snow depth on ground)? p.11 Section 4.2 This is a very minor point. Here the authors present the results from the secondary decision trees that split the dataset into high or low expressions of the six key variables. The results are presented in Table 5. The results start with a discussion of high and low snow cover years and then moves on to high and low total snowfall. These two variables are presented in reverse order in Table 5. You might want to change either the Table or the results section. p.12. line 5 The authors describe that the top three controls on runoff ratio matched the overall hierarchy for high snow cover years, albeit with differing orders of importance. Maybe you could quickly include that changed order so that the reader does not have to look for Table 5 to find out what that order was. p.12 line 11 The authors state that for low fall soil surface water content the analysis indicated that mean daily wind speed had a large influence on runoff ratios. Could you maybe discuss why this could be the case. The connection is not clear to me at all. p.16 Figure 4 This Figure is very difficult to understand. It needs to be explained in much more detail. Also the inserted panel in the right hand figure showing Figure 3.3. (why 3.3?) is way too small to be readable. I would recommend removing it entirely. The left hand panel of the Figure is not explained in the text at all and the rudimentary legend box is not enough to make it understandable. P17. Line 29 The authors state once again that the six key variables explain most of the variability and exert the greatest control on runoff ratio. Maybe one could add a discussion here of whether including only these six variables is "good enough" for a model or how much additional information the other 9 used parameters really provide. P18 Figure 6 Intuitively I would have expected the wet and dry scenarios in panel D to be presented in different order, i.e. wet on top and dry on the bottom. p.18 end This sentence is virtually identical to a sentence on p. 16 lines 12 and following p. 20 line 14, 15 If the relationship was so strong, why wasn't length of frozen soil over the winter not included as a predictor? Please add explanation.
* * *

---

## Author Comment (AC1) · 16 Feb 2017

Author responses to Reviewer #1's interactive comments on: Hydrol. Earth Syst. Sci. Discuss., doi:10.5194/hess-2016-564, 2016 "The hierarchy of controls on snowmelt-runoff generation over seasonally-frozen hillslopes" by A. E. Coles et al.

Authors:

Summary

Thank you to the three reviewers (Reviewer #1, Reviewer #2, and S. Schälchli) for their in-depth reviews of this manuscript. Your reviews are very useful in improving this manuscript. We reply to each of the comments in separate documents. But we wanted to provide a brief summary here of what we saw to be the six main comments on the

manuscript (some highlighted by more than one reviewer) that we will improve in the revised version:

1. Literature review. We will include more descriptions of what existing knowledge tells us about how each of the predictor variables individually influences runoff. We will extend our literature review to outside of the Canadian Prairies.

2. Transferability of findings. We will add more of a discussion on how these findings are transferrable to other regions.

3. Methodology. We will add more description of how the decision tree elements can be interpreted. We will make the methodology section less technical.

4. Predictor variable correlation. We will remove some of the predictor variables that correlate with others, and restructure the decision tree based on that.

5. Land cover and tillage. We will add tillage to the decision tree analysis, and add more discussion about land cover and tillage effects on runoff.

6. Implications for modelling. We will provide more discussion on specifically how this decision tree analysis can improve modelling approaches, such as statistical 'add-ons' to existing empirical approaches.

We talk about these six aspects in our responses, as well as several other comments that the reviewers had.

Below are the comments from Reviewer #1 and our responses to those comments.

Reviewer #1: The manuscript aims at identifying the relative importance of controls on snow-melt runoff from three arable plots of approximately 5 ha located in the Canadian Prairies. To this end they use a statistical method – decision trees – with a set of 15 predictor variables to analyze nearly 50 years of runoff data. The authors conclude that the relative importance of controls is not general for all winters, but changes (for example) with the thickness of the snow cover. As the authors correctly show in the

introduction, there has been a substantial amount of experimental studies in the past 30 years on winter-time runoff generation with a focus on the snow cover and the frozen soil – at different scales (from small plots to catchments) and in different regions of the world (incl. Canadian Prairies, Northern Scandinavia, Alpine areas and Japan). So, I dare say that we know quite a lot already about important controls of snow-melt infiltration and runoff – including formation and permeability of a frozen soil layer.

Therefore my first reaction when I started reading this manuscript was: do we really need a statistical analysis to find out the "hierarchy of controls" on snowmelt-runoff for seasonally frozen areas? And, what's the added value of such a statistical analysis to the existing knowledge from emprical studies and deterministic modelling? After having finished reading the manuscript I have the feeling that the lessons learned from this exercise are marginal.

Authors: Thank you for your comments. It is unfortunate that you feel that the lessons learned are marginal and that we haven't properly detailed the added value of this statistical analysis. We do think this analysis has provided added value. We agree with you that we already know a lot about the important controls on snowmelt-runoff, and we refer to these up front with previous research in the Introduction section. We will rework the literature review to better summarise what is already known mechanistically about the important predictors on snowmelt-runoff (while also expanding this to other, non-Prairie studies on snowmelt-runoff).

Despite our existing in-depth knowledge, however, there are persistent problems with modelling snowmelt-runoff. This is likely down to the fact that runoff generation and runoff ratios are strongly spatially and temporally variable. At one site, the runoff ratio of the springtime snowmelt could vary hugely from one year to the next. While individual controls and their influence on runoff are well understood, we believe that this high variability in runoff is in part due to interactions – sometimes subtle – between controls, and shifting relative importance depending on other conditions. We believe that teasing apart these interactions and hierarchies of controls – especially

the condition-dependent hierarchies – is the first step in designing more effective monitoring schemes, and incorporating the right parameters together in empirical models.

Reviewer #1: This has to do with a number of critical drawbacks of this study: a) The number of winter runoff situations (response variable) is critically low compared to the large amount of predictors (15).

Authors: For the three hillslopes, we have 140 runoff ratios between 0 and 1. For each of these runoff ratios we had a set of 15 predictor variables. This is enough observations to allow for a decision tree approach. As explained in the manuscript, we believed it be overly pedantic and physically unrealistic to attempt to predict runoff ratio values (e.g. RR=0.55 given a set of predictors). It was more appropriate to group the runoff ratio responses, and we settled on 5 classes for this, and instructed the decision tree model to predict the runoff ratio class.

We agree that the large number of predictor variables means that the approach is at risk of model over-parameterisation and overfitting. However, we took steps to avoid this by using a technique called pruning. This meant that not all predictor variables (only 2 - 6) were used in any one decision tree.

Reviewer #1: b) Some of the predictor variables are highly correlated with other predictor variables.

Authors: Yes, we ran a correlation analysis. Snowfall and snow cover are significantly correlated ($p<0.05$), as are fall soil surface water content and fall soil profile water content. You're right that they both should not be included in the analysis. We will remove snowfall and fall soil profile water content.

Reviewer #1: c) We know from many field experiments that runoff in spring is not the result of average winter conditions, but can reflect critical short situations of the winter; e.g. short midwinter melt events, or short coincidence of a shallow snow pack and very cold air temperatures generating substantial soil frost. This applies in particular to

regions with a broad range of soil frost conditions, such as the Canadian Prairies, that are sensitive to the snow-cover thickness. In conclusion, an analysis based on important predictor variables representing mean winter conditions (e.g. mean temperature or total seasonal snowfall) is probably not very conclusive.

Authors: We agree! We are limited somewhat since each predictor variable must constitute only one value for each response variable (seasonal runoff ratio), therefore some conditions can only be captured by averaging or summing over a period. However, the only predictor variable that used a general winter condition was 'total seasonal snowfall'. Our other winter-related predictors attempted to get at these critical short situations of the winter. For example, 'mean temperature on warm winter days' is not mean winter temperature but instead is the mean temperature of days which had air temperature that exceeded 0°C AND coincidentally had snow cover on the ground (i.e. an indicator of potential over-winter snowmelt event). A second example is the 'mean daily wind speed above blowing snow threshold', which is the mean wind speed for days where winds exceeded 7.5 m s-1, which is the threshold for blowing snow redistribution. Thus, these are attempts to obtain metrics of these specific conditions, while encapsulated in a single value, and aren't just a generalised winter condition.

Reviewer #1: There are a few other issues with this study that I consider as critical: - The title and some parts of the manuscript imply that the relative importance of the different controls on snowmelt-runoff generation – found in this study – are general. But in fact, it only applies to the specific slope, soil and meteorological conditions of this field site. How can the hierarchy of controls found at this site be transferred to other sites with other snow and soil conditions, or with another topography?

Authors: You are right in that the analysis is based solely on conditions at this study site. We will refocus those parts of the manuscript that inappropriately imply that they are general to all snowmelt-runoff systems. We will also add a discussion section on transferability to other sites. Although only one site with three hillslopes is studied here, we believe that it enables transferability since, with such a long dataset, we have observa-

tions of multiple snow and meteorological conditions, multiple soil moisture conditions, multiple frozen ground conditions, and three topographic realisations. So, in a sense we are substituting space for time. We would suggest that limits to transferability are the soil type (silt loam) and agricultural nature, since these are two aspects that are unchanging through the study period.

Reviewer #1: - The reader gets very little information about the specific conditions of this site. In fact, it seems that what is called "snow-melt runoff" is only "surface runoff". What about lateral discharge? Is there a groundwater table fluctuating near the surface? What do we know about the soil type and the soil physical properties? How does a typical snow cover look like in this area? I guess it must be very patchy and loose. What's the typical length of the winter season? Such information would be essential for interpreting the results.

Authors: Thank you for pointing out that we need more site-specific information. We will include much more detailed information in Section 2 (Study site and dataset) concerning the aspects that you point out. Briefly here, yes runoff from the hillslopes is from surface runoff. The groundwater table is several meters below the soil surface, and does not fluctuate near the surface. The soil is a silt loam (we already state this in the manuscript), with very low infiltration capacities when frozen (0.09 to 2.57 mm hr-1, which we observed in laboratory experiments). There might be small amounts of lateral flow after thawing, but it is reasonable to assume that is unimportant. We will also include bulk density, soil depth, and unfrozen infiltration capacity data. No, snow cover is not patchy and loose. Typically, the site is 100% snow-covered over winter and into the spring snowmelt season. End of winter hillslope-averaged snow depth is up to 40 cm (much deeper in topographic depressions, shallower on exposed areas), while snow density averages 0.24 g cm-3 and can be up to 0.45 g cm-3. The winter season is typically 4-5 months in duration, when up to 150 mm precipitation falls as snow. We will incorporate all of this information into the revised manuscript.

Reviewer #1: - Two of the most important predictor variables are "Total seasonal snowfall" and "snow cover (i.e. peak SWE)" (see Fig. 3) – (which is by no means a surprise) – and I assume that these two variables are highly correlated. How do the authors justify the selection of these two predictor variables?

Authors: Yes they are correlated. We had previously included both since the decision tree's selection of one would tell us which held more predictive power. However, as mentioned above, we now suggest that we should remove snowfall due to the correlation. In its place, we would bring in a new variable that indicates over-winter ablation (e.g. the difference between snowfall and snow cover).

Reviewer #1: - One key-variable for snowmelt-runoff generation at this site is completely missing in the analysis (and also in the discussion!): the extent of the soil frost. And with that I mean both the frost depth and the ice content. I didn't find any information about typical frost depths, for example. Of course, there is an implicit account of soil frost because of the strong (negative) relationship between snow cover and soil frost depth. And there is an implicit relationship between pre-winter soil water content and ice content later in the winter. But this obvious key-connection is not made.

Authors: We disagree that it is missing! We included what we called 'thawed layer depth' as a predictor variable, which is the depth between the soil surface and the top of the frozen layer on the peak runoff day. It is an indicator of whether runoff was occurring over ground frozen right to the soil surface, or over ground that was thawed. Typically, the ground is frozen to the soil surface ('thawed layer depth' = 0 cm) at the time of peak runoff. Given that this system is a surface runoff-dominated flow regime, this measure of the upper boundary of frozen ground is important. You are right that the pre-freeze soil water content (fall soil water content) gives an indication of the frozen ground ice content. Apologies if this connection was not made clear in the manuscript; we will make sure that it is clear in the revised manuscript. We do actually discuss at length the reasons why frozen ground ice content might vary from pre-freeze soil water content (most notably in Section 5.2 on 'Soil moisture memory'). As for the soil profile depth to which frozen conditions typically extend (the lower boundary of frozen

ground), we will add information on typical values for this in the study site (along with the information on soil depth, bulk density, infiltration capacity, etc.). We also discussed observations that longer periods of deep soil freezing (at depths of 50 cm and 100 cm) is positively correlated with runoff ratio in the following spring (page 20 lines 14-21). We will run a correlation analysis to assess the correlation between the depth of frozen soil at the time of runoff and snow cover depth (SWEc). If there is no significant correlation, we will include this as a predictor in a revised decision tree analysis.

Reviewer #1: - How about land-use? According to chapter 2 the three plots were covered with different vegetation and experienced different tillage practices. But it seems that this was not of major importance for the snowmelt-runoff generation. (Also the topography of the three slopes is more or less the same.) If that's the case, I think that the decision tree actually has only an N of 52 (and not 140), which makes the statistical analysis even more questionable. However, if vegetation and tillage have a significant impact, then it should be also discussed somewhere.

Authors: We described findings that showed fallow hillslopes had generally higher runoff ratios than other land cover types (page 15 line 25-27). We also outlined results of the effect of standing stubble on snow retention, and its potential knock-on effects on soil water content and runoff ratios (page 15 line 27 onwards). We included land cover variable in the decision tree analysis, but it was rarely (just once) picked out as an important control on runoff. We suggest that this is because soil moisture and snow cover are the avenues by which land cover influences runoff (vegetation effects on soil moisture depletion, and snow-trapping effects from crop residue), and therefore it is those factors that are picked out by the decision tree rather than land cover directly. We can include more discussion on the role of vegetation, and also include tillage in the analysis, in the revised manuscript.

Reviewer #1: - Chapter 5.3: Implications for modeling and future field campaigns: are we going to change current practices in modelling snow-melt runoff (in regions with seasonally frozen soil) based on the lessons learned from this statistical analysis? I

don't think so. The key-role of the snow cover and the significance of pre-winter soil moisture content has been known for quite a long time and is accordingly represented in current runoff models. And also the critical need for accurate and spatial snow-cover data is well recognized.

Authors: There are several empirical approaches that estimate infiltration into frozen ground that use two parameters: soil moisture and SWE. We tested one of these – the Granger equation, which is a widely-used model – and showed that the two-parameter model does not work as well as the approach that we have presented in this paper. While, yes, the influence of the individual controls has previously been shown extensively, here we were able to show with a large dataset the interactions and hierarchies not usually apparent.

We propose that we will cut down the paragraph on page 22 lines 11-26 to make it more concise. We will add discussion on how existing empirical approaches can be improved with the findings of this study. One option could be to design an error model based on the Swift Current data that accounts for the variability captured by these extra parameters, and apply it as an add-on to the existing Granger empirical equation.

Reviewer #1: - Finally, the world-wide knowledge on snowmelt-runoff generation in areas with seasonally frozen soil is not well reflected in the introduction. Only Canadian studies are referred to.

Authors: Yes, you're right. We will incorporate more studies from outside of the Canadian Prairies.

Reviewer #1: In conclusion, I'm not convinced that the above-mentioned problems can be solved with the available data and the selected method to become a contribution that adds value to the existing state-of-the-art knowledge on snowmelt-runoff formation in cold regions.

Authors: We believe that we have addressed the comments that you have made. We

agree that more work can be done to improve this manuscript, and we know the ways we will do this. We think that this rare, long-term, multi-variable dataset and the decision tree method do actually contribute a great deal to extending our knowledge on the interactions, feedbacks and condition-dependent hierarchy of controls on runoff.

———————————————

---

## Author Comment (AC2) · 16 Feb 2017

Author responses to S. Schälchli's interactive comments on: Hydrol. Earth Syst. Sci. Discuss., doi:10.5194/hess-2016-564, 2016 "The hierarchy of controls on snowmelt-runoff generation over seasonally-frozen hillslopes" by A. E. Coles et al.

Authors:

Summary

Thank you to the three reviewers (Reviewer #1, Reviewer #2, and S. Schälchli) for their in-depth reviews of this manuscript. Your reviews are very useful in improving this manuscript. We reply to each of the comments in separate documents. But we wanted to provide a brief summary here of what we saw to be the six main comments on the

manuscript (some highlighted by more than one reviewer) that we will improve in the revised version:

1. Literature review. We will include more descriptions of what existing knowledge tells us about how each of the predictor variables individually influences runoff. We will extend our literature review to outside of the Canadian Prairies.

2. Transferability of findings. We will add more of a discussion on how these findings are transferrable to other regions.

3. Methodology. We will add more description of how the decision tree elements can be interpreted. We will make the methodology section less technical.

4. Predictor variable correlation. We will remove some of the predictor variables that correlate with others, and restructure the decision tree based on that.

5. Land cover and tillage. We will add tillage to the decision tree analysis, and add more discussion about land cover and tillage effects on runoff.

6. Implications for modelling. We will provide more discussion on specifically how this decision tree analysis can improve modelling approaches, such as statistical 'add-ons' to existing empirical approaches.

We talk about these six aspects in our responses, as well as several other comments that the reviewers had.

Below are the comments from S. Schälchli and our responses to those comments.

S. Schälchli: Review

The article includes many very interesting contents and also adds new results to the broader scientific context. Furthermore they came up with new ideas and used tools, which were never used in this research context before. The Title reflected the contents of the paper clearly.

In the beginning the article was quite hard for me to read. I had to read some paragraphs twice and look up some technical terms. Especially the abstract uses, from my point of view a quite difficult language and the theme often jumps among the different results.

Authors: Thank you for your review and helpful comments. We will make sure all technical terms are defined. We will edit the abstract to improve the flow of results.

S. Schälchli: The Introduction was written clearly and the authors made a good assumption of the present scientific stage. Furthermore they linked their research to other papers and declared why their research topic is of relevance. They generated a very good embedding of their topic into the broader scientific range. The research questions were pointed out clearly. But the 3rd Question was answered insufficient, because the author didn't mention in what way the hierarchy varied from year to year.

Authors: You're right. We neglected to update the terminology of the research questions. We will edit 'year to year' to 'condition-dependent' variability in hierarchy.

S. Schälchli: The used data was in the majority good, but as they also made clear, the measurements of the soil water content were made a bit too seldom. The measurements have been done in October (before the freeze-up) and then only in April when the melting process has already begun in many years. This is not an ideal presupposition for the correctness of the soil water content and therefore for the whole influence of the soil moisture on the runoff output. The reason is that until April there may occurred infiltrations (rising of soil water content) or percolations (decrease of soil water content).

Authors: Yes, we discuss this in the paper. The soil water content measurement in the spring was always carried out after the end of the spring snowmelt season. Therefore it absolutely does reflect any change in soil water content over winter (increases due to mid-winter ablation events or upward migration towards the freezing front, or decreases due to deeper percolations) as well as increases during the spring snowmelt season.
We detail this clearly. It is unfortunate that we do not have pre-melt soil water content data. A recommendation of this paper is the importance of observing pre-melt soil water content to account for any over-winter changes in the soil water content.

S. Schälchli: Another question crossed my mind when I read the chapter 2 on the study side and the dataset. Is there maybe a difference in the snow adhesion by the diverse crop types stubbles? For example in term of the stability respective to wind redistribution?

Authors: Yes, we show that on p.15 line 29-31 that wheat stubble (which occurs when a wheat crop year coincides with no tillage in the fall, to leave standing residue) retains more snow cover than, for example, fallow, due to the snow-trapping qualities of standing stubble and less redistribution by wind. While other vegetation (grass, lentils) do leave 'residue' on the hillslopes if not tilled, they do not have the same snow-trapping qualities as standing wheat stubble.

S. Schälchli: I have to point out that I wasn't completely able to follow up with the methodology part. For example it was unclear to me how the authors came up with the decision in which hierarchy the controls are in an explicit case and I also had my difficulties to read the decision tree. First when I read the whole paper trough and then returned to the abstract or to the decision tree the intentions became clearer.

Authors: Thank you for that feedback. Reviewer #2 also found the methodology and decision tree elements difficult to understand. We will work on the methodology section and improve its clarity, with focus on how the hierarchy was decided and how the decision tree elements can be interpreted.

S. Schälchli: The results and the conclusions were written very comprehensible and contained many good visualisations and graphics. The Figure 4 and 5 were in particular very illustrative and informative. However why there is a sharp step of the seasonal snowmelt-runoff at 70 mm is unclear to me. Their results were highly interesting, especially in terms of the influence of the soil water content under the seasonal frozen

conditions on the runoff output. I support their appeal to measure also the pre melt soil water content (Coles, A. E. et. al. (2016): 23). Than during reading the results I was thinking about the faults which may occur when the soil water content is only measured twice a year and thus was very happy to find my thinking's reinforced in the conclusion part by the authors themselves.

Authors: Thank you for your feedback about the source of the 'step' in Figure 4 being unclear. The step is at 71.5 mm for total seasonal snowfall (x-axis). We incorporated that into this figure because the decision tree (Figure 3) found that total seasonal snowfall, with this threshold amount (71.5 mm) to be the first control (the first node in the tree) on determining runoff ratios. Since Figure 4 was a partitioning of the relationship between total seasonal snowfall and total seasonal snowmelt-runoff, we thought it made sense to break it up according to the first branching of the decision tree. However, we appreciate that this is a very definite line, while the partitioning ought to be more 'fuzzy' given the <100% predictive accuracies and multiple combinations of variables that dictate a runoff ratio. We will edit this figure to remove this step, and exhibit a more 'fuzzy', less definite partitioning of the relationship between snowfall and snowmelt-runoff.

S. Schälchli: Finally the conclusions were hold shortly and precise with a clear statement addressing further research on the influence of soil water content in seasonal frozen soils on the runoff output.

Authors: Thank you. We're happy you found them to be precise and clear.

S. Schälchli: Recommendation List (bold = major points, italics = minor points): • Central theme in abstract have to be found and a better organisation of the results (in Abstract) should be done • 3rd Research question: Variations in between of the different years were not addressed • Method part should be better clarified and the functioning of the decision tree should be explained better • Figure 4: Author should explain the occurring sharp step at 70 mm in the manuscript. And are the shapes in the soil boxes the granularity of the soil or ice lenses? Label it clearer. • Comparison of

the different crop types and their effect on snow cover and runoff • Interesting would be how the determining factors changed over time and if there is a trend in the data (maybe caused by climate change?). If there is a trend: How does the change in the determining factors influence the snowmelt runoff?

Authors: We have addressed most of these comments in our responses above, and will edit the manuscript accordingly. We will edit Figure 4's legend to make that clearer. Regarding the trends in the data, long-term climate and runoff trends at this site was the subject of another study (Coles et al., 2017), which we will refer to in the next version of this manuscript.

Thank you again for your review of our manuscript!
* * *

---

## Author Comment (AC3) · 16 Feb 2017

Author responses to Reviewer #2's interactive comments on: Hydrol. Earth Syst. Sci. Discuss., doi:10.5194/hess-2016-564, 2016 "The hierarchy of controls on snowmelt-runoff generation over seasonally-frozen hillslopes" by A. E. Coles et al.

Authors: Summary

Thank you to the three reviewers (Reviewer #1, Reviewer #2, and S. Schälchli) for their in-depth reviews of this manuscript. Your reviews are very useful in improving this manuscript. We reply to each of the comments in separate documents. But we wanted to provide a brief summary here of what we saw to be the six main comments on the manuscript (some highlighted by more than one reviewer) that we will improve in the revised version:

1. Literature review. We will include more descriptions of what existing knowledge tells us about how each of the predictor variables individually influences runoff. We will extend our literature review to outside of the Canadian Prairies.

2. Transferability of findings. We will add more of a discussion on how these findings are transferrable to other regions.

3. Methodology. We will add more description of how the decision tree elements can be interpreted. We will make the methodology section less technical.

4. Predictor variable correlation. We will remove some of the predictor variables that correlate with others, and restructure the decision tree based on that.

5. Land cover and tillage. We will add tillage to the decision tree analysis, and add more discussion about land cover and tillage effects on runoff.

6. Implications for modelling. We will provide more discussion on specifically how this decision tree analysis can improve modelling approaches, such as statistical 'add-ons' to existing empirical approaches.

We talk about these six aspects in our responses, as well as several other comments that the reviewers had.

Below are the comments from Reviewer #2 and our responses to those comments.

Reviewer #2: The authors use a decision tree method and a 52-year data record of snowmelt related runoff from three agricultural plots in the Canadian prairies to study the controlling factors on runoff generation over seasonally frozen soils. Spring runoff conditions are characterized by determining runoff ratios. Decision trees are then used to understand the relative importance of a number of control parameters describing topography, land use, vegetation, and precipitation dynamics on the spring runoff. The intent of the study is to improve the understanding of the relative importance of the factors responsible for runoff over frozen grounds which should help to improve models describing infiltration into frozen ground and runoff over frozen hillslopes. The authors

also want to give some advice to future measurement campaigns looking at seasonally frozen, snowmelt dominated regions. The authors claim that the major innovative part of their study as compared to other studies looking at runoff generation cover frozen ground is, that those studies usually are based on short term experiments and single season runoff events, while their study looks at averages over a much longer data period. This longer record should help to reveal the nonlinearities and interactions between the various process controls. In my opinion, this aspect of the study could indeed present some important new insights into the scientific field of snowmelt runoff generation over frozen ground. The paper is generally well written and the language and citing are appropriate. However, I find parts of it (especially in the Methods section) rather difficult to follow (see comments). The introduction and literature review focuses very narrowly on studies mostly from western Canada. While this is understandable due to the study location, I think a broader look at studies of runoff over frozen soils would be beneficial. The research goals on the other hand are clearly stated and well defined. The results and discussion section are adequate but could maybe be structured a little more clearly (see comments). The conclusions are based well on the results and provide a good summary of the study. The Tables are concise and easy to interpret, while some of the Figures could be improved (see comments) to make them easier to grasp. The topic of the article falls within the scope of the journal and does in my opinion represent a worthwhile contribution to the snow science community. However, the article needs in my opinion some fairly substantial changes to make it less difficult to follow. I would therefore recommend publication after major revisions (see general and specific comments).

General Comments

As mentioned, the literature review in the introduction should be broadened to include at least some studies of runoff and runoff generation over seasonally frozen or even permafrost soils. I would also welcome a discussion of the relevance of the studies result for other regions of the globe. Are the study results only applicable to prairie

landscapes or could similar hierarchies and interactions be found for example in high arctic locations that have frozen soils and often similar topography. This discussion could also be included towards the end of the paper in the general discussion section.

Authors: Thank you for your review of this manuscript. We appreciate your comments. We agree that we did not provide an adequately expansive review of literature on runoff over frozen ground, especially in regions outside of the Canadian Prairies. We will add a more comprehensive literature review in the revised manuscript. Yes – we believe the study results are applicable beyond the Canadian Prairies – to other gently sloping, frozen-ground environments of North America and northern Eurasia, but likely limited to similar agricultural, soil types. We will add a discussion section on transferability.

Reviewer #2: I found it somewhat hard to follow the explanation of the decision tree method in the Methods section. The description is very technical, which makes it hard for the average reader to understand what the analysis really means. I think it would be beneficial to include some explanations as to what it means f.e. if some predictors occur several times in a tree, what the ranking of the nodes mean for the importance to runoff generation, and what a leaf really represents "in reality". I guess I would recommend to "dumb down" the explanation a little bit so that readers who are not familiar with the decision tree technique can appreciate what the single tree elements (nodes, leafs etc.) mean for the description of the "real life" process.

Authors: It is useful to hear this, thank you. We went back and forth in old versions of this manuscript with a technical vs. 'dumbed down' explanation of decision tree methodology and tree elements. We settled on a shorter, more-technical version following feedback from others that the methodology section was too long and we were over-explaining a 'logical' process. However, after hearing from you and another HESS reviewer that the methodology is hard to follow, we will revert to a more accessible and clearer explanation of decision trees. We will edit this in the revised manuscript.

Reviewer #2: I would really encourage a much more in depth presentation and discussion of the predictor variables as part of the discussion. The choice of which predictor variables to use is a major point of the study and should be treated as such. So far information on the predictors is only available within Table 2 and is very short. I would especially welcome a short description for each predictor, why they were chosen, and how they theoretically are expected to influence runoff. Additional points that could be discussed here are: What processes are believed to lead to the differences between total snowfall SWEf and snow cover SWEc. How closely are these two predictors related (maybe include an autocorrelation analysis). What impact does the soil fall water and fall soil profile water content have on the mechanics of spring infiltration (i.e. formation of concrete ice soils when water content is high, this might not be obvious to readers not that familiar with frozen soil hydrology)? Why is mean daily wind speed needed? The process deposition or removal of snow from the test plots due to blowing snow should be included in the difference between total snowfall SWEf and snow cover SWEc. So what additional information or impact does blowing snow during the winter have on spring runoff. The authors use 4 predictors to describe melt conditions. Wouldn't melt season length and melt rate be enough? What additional information do the other two predictors provide?

Authors: Absolutely. We will include further information on the predictor variable choices and how they theoretically, based on existing literature, should influence runoff.

The differences in total snowfall SWEf and snow cover SWEc are likely driven by over-winter redistribution by blowing snow and any over-winter melt events due to increased temperatures. As described in the comments above, because of the correlation between snowfall and snow cover, we will remove snowfall from the decision tree analysis in the revised manuscript.

Regarding a discussion of the impact of fall soil water content on the mechanics of spring infiltration, yes we will make this clearer in the revised manuscript and refer more to fundamental principles in frozen soil hydrology. We will include explanation of the effects of the soil moisture itself (like in unfrozen soils), but also temperature factors

such as heat transfer, melting and re-freezing, movement of water down temperature gradients, freezing of infiltrated water, and pore blockages by ice, etc.

We included mean daily wind speed above the blowing snow threshold as a predictor that might account for over-winter ablation of the snow cover (blowing snow and sublimation together can transport and sublimate up to 75% of annual snowfall from open, exposed fallow fields; Pomeroy et al., 2010). In other work by our group (Coles et al., 2017), we have shown that blowing snow is the main ablation mechanism when the hillslopes were in fallow, while over-winter temperature is the main ablation mechanism when the hillslopes had standing stubble. We will make reference to this effect in the discussion of the revised manuscript. Further, non-sublimated blowing snow forms drifts, which influence the streamflow peak and duration, which might in turn have knock-on effects on runoff ratio (Pomeroy et al., 2010).

Regarding the four predictors to describe melt conditions, we believed that they each provided useful information. Apart from melt rate and melt season length, we also included date of peak runoff and spring temperature gradient as predictor variables. Date of peak runoff is a useful measure as it provides information about the timing of spring runoff in the year: for example, whether it was early spring or late spring, which anecdotally has been thought to relate to the rapidity of melt, and infiltration and runoff amounts. We included spring temperature gradient as another indirect indicator of melt rate. You are right in suggesting that it was superfluous, and we can remove it from the analysis in the next manuscript revision.

Reviewer #2: Specific Comments

p.4 Line 5 and following The authors describe the usual conditions on the test plots as they were implemented in the decision tree analysis and the exceptions to those conditions. Unfortunately, considering the study period of 52 years there seem to be quite a lot of extraordinary conditions considering land use and tillage. One would assume that especially different tillage practices and the presence of stubble or standing crop

over the winter (additional catch of blowing snow as mentioned later in the paper) could have quite an impact on runoff. Yet it seems like all years were included in the analysis. At the very least I would expect a discussion of the possible impacts of this.

Authors: Actually, this comment isn't entirely accurate. We didn't only implement the usual plot conditions into the decision tree analysis and ignore the exceptions. We included all conditions in the analysis: the decision tree analysis directly includes the land covers for each plot and each year (whether 'usual' or 'exceptions'). Further, we don't agree that the exceptions amount to "quite a lot"; the exceptions to the usual wheat-fallow land cover rotation amount to only 16% of the occurrences, and the exceptions to the usual conventional tillage practice amount to only 11.5% of the occurrences. But again, we did differentiate between these in the case of land cover. We did not include tillage differences in the decision tree analysis. This was partly because previous work (Coles et al., 2017) had shown that long-term runoff trends are similar across all three hillslopes (no difference in the magnitude or significance of trends, despite the fact that the tillage practice on one of the hillslopes - Hillslope 2 – had been changed from conventional to zero-till in the last 18 years of the record), which might suggest that tillage changes have not played a role in the runoff regime. We will refer to this study in the revised manuscript. However, theoretically, tillage would affect runoff ratios due to influence on soil structure, macropore development, surface water retention and infiltration. We can therefore explicitly include this variable in the revised manuscript.

Reviewer #2: p.6 line 25 The runoff ratio is defined as total runoff divided by SWEf from the end of the hydrological year to the end of snowmelt. How was liquid precipitation (i.e. rain) handled in this? I'm aware that in Saskatchewan rain after the end of the hydrological year and before the start of the snowfall and during snowmelt is rather rare, but it must have occurred sometimes during the 52 year study period. Also how was SWEf measured at the met station (precipitation sampler, snow depth on ground)?

Authors: Rainfall during this time period was indeed rare, yes. We used an all-weather Belfort weighing gauge (information given on p.5 line 11) and distinguished between

snowfall and rainfall using the air temperature data (we will add this method of distinguishing between snowfall and rainfall to that section). We summed all snowfall instances to give total seasonal snowfall, SWEf (mm).

Reviewer #2: p.11 Section 4.2 This is a very minor point. Here the authors present the results from the secondary decision trees that split the dataset into high or low expressions of the six key variables. The results are presented in Table 5. The results start with a discussion of high and low snow cover years and then moves on to high and low total snowfall. These two variables are presented in reverse order in Table 5. You might want to change either the Table or the results section.

Authors: Good point, we will reverse the order in Table 5.

Reviewer #2: p.12. line 5 The authors describe that the top three controls on runoff ratio matched the overall hierarchy for high snow cover years, albeit with differing orders of importance. Maybe you could quickly include that changed order so that the reader does not have to look for Table 5 to find out what that order was.

Authors: Good point, we will add this in for easy-reading.

Reviewer #2: p.12 line 11 The authors state that for low fall soil surface water content the analysis indicated that mean daily wind speed had a large influence on runoff ratios. Could you maybe discuss why this could be the case. The connection is not clear to me at all.

Authors: Blowing snow creates drifts, which, on these Swift Current hillslopes, are in the immediate vicinity of the runoff flumes. This might affect runoff ratio due to the drifts' influence on streamflow peak and melt duration. There might also be a statistical reason for the selection of mean wind speed in the decision tree model. We will discuss this further in the revised manuscript.

Reviewer #2: p.16 Figure 4 This Figure is very difficult to understand. It needs to be explained in much more detail. Also the inserted panel in the right hand figure showing

Figure 3.3. (why 3.3?) is way too small to be readable. I would recommend removing it entirely. The left hand panel of the Figure is not explained in the text at all and the rudimentary legend box is not enough to make it understandable.

Authors: Thanks for this feedback on this figure. The inserted panel is a small version of Figure 3 (apologies for it saying "Figure 3.3", this was an error) and is not intended to be readable, but solely to remind the reader that Figure 4 and its colour-coding are dictated by the findings and colour-coding of Figure 3. However, we appreciate that it might be confusing to have this insert here at all, and will trust that simply stating in Figure 4's caption that it is "colour-coded using leaf colours in Figure 3" is enough and we therefore do not need the insert. We will remove the insert. Figure 4 is an attempt to return to the scattered relationship between precipitation and runoff (Figure 2) and account for this scatter using the findings of the decision tree (Figure 3; Tables 4-6). We state this in Figure 4's caption and in the main text at p.16 line 9-10. However, we will add further text to make this clearer. We will also improve the legend (which is showing the principal conditions that drive the runoff response in each part of the partitioned scattered figure) to make that clearer.

Reviewer #2: P17. Line 29 The authors state once again that the six key variables explain most of the variability and exert the greatest control on runoff ratio. Maybe one could add a discussion here of whether including only these six variables is "good enough" for a model or how much additional information the other 9 used parameters really provide.

Authors: We stated in the manuscript that those six variables explain 70% of the runoff ratio variance. Whether that is 'good enough' for modelling is likely subjective and depends on the desired performance of the model. Compared to the 14.3% of the runoff ratio variance that is explained by an existing empirical modelling approach (discussed in section 5.3), this 70% is a vast improvement. We think that including 6 parameters in a model that has 140 observations is already a lot, and including any or all of the remaining 9 would risk over-parameterisation. We will add a small discussion in this

section about this. Reviewer #1 wondered about what realistic modeling improvements we could offer from the findings of this study. We suggested a statistical, error-term type add-on to existing empirical two-parameter (soil moisture and SWE) approaches that would account for the variability described by the additional four parameters found here.

Reviewer #2: P18 Figure 6 Intuitively I would have expected the wet and dry scenarios in panel D to be presented in different order, i.e. wet on top and dry on the bottom.

Authors: Yes, that would be better. That way all 'high runoff ratio' examples are on the top row, and 'low runoff ratio' examples are on the bottom row. We will change this.

Reviewer #2: p.18 end This sentence is virtually identical toa sentence on p. 16 lines 12 and following

Authors: Yes, it is the same point. However, that was intentional. The paragraph on p.16 lines 5-13 is the start of the discussion section, and is a summary of the discussion that is to follow in the next sub-sections (highlighting the main results that will be discussed). We would prefer to keep it like this.

Reviewer #2: p. 20 line 14, 15 If the relationship was so strong, why wasn't length of frozen soil over the winter not included as a predictor? Please add explanation.

Authors: This was later analysis that was carried out after the decision tree analysis in a direct attempt to explain why sometimes soil moisture memory appears to be strong and other times not. It was analysis based on a learning outcome and discussion of the decision tree findings, and was not considered prior to the decision tree analysis. As mentioned above in response to a comment by Reviewer #1, we will include a measure of frozen ground depth at the time of runoff, into the decision tree analysis.

References

Coles, A.E., McConkey, B.G., and McDonnell, J.J. (2017) Climate change impacts on hillslope runoff on the northern Great Plains, 1962-2013, Journal of Hydrology, in review.

Pomeroy, J., Fang, X., Westbrook, C., Minke, A., Guo, X., & Brown, T. (2010) Prairie Hydrological Model Study Final Report. Saskatoon: University of Saskatchewan.